# LaCy: What Small Language Models Can and Should Learn is Not Just a Question of Loss

## Abstract

Language models have consistently grown to compress more world knowledge into their parameters, but the knowledge that can be pretrained into them is upper-bounded by their parameter size. Especially the capacity of Small Language Models (SLMs) is limited, leading to factually incorrect generations. This problem is often mitigated by giving the SLM access to an outside source: the ability to query a larger model, documents, or a database. Under this setting, we study the fundamental question of *which tokens an SLM can and should learn* during pretraining, versus *which ones it should delegate* via a `<CALL>` token. We find that this is not simply a question of loss: although the loss is predictive of whether a predicted token mismatches the ground-truth, it is insufficient for identifying which predictions would actually lead to factual or semantically invalid continuations. Some high-loss tokens correspond to *acceptable* alternative continuations of a pretraining document and therefore should not trigger a `<CALL>`. This suggests that learnability cannot be characterized from loss alone, but requires additional domain-specific signals about the role of a token in the sentence. In Wikipedia-like domains, we show that augmenting the loss signal with lightweight grammatical information from a spaCy parser substantially improves delegation decisions. Based on this insight, we propose LaCy, a novel pretraining method that combines loss with factuality signals to decide which tokens an SLM should learn. Our experiments demonstrate that LaCy models successfully learn which tokens to predict and when to call for help. This results in higher FactScores when generating in a cascade with a bigger model and outperforms Rho or LLM-judge trained SLMs, while being simpler and cheaper.

## 1 Introduction

Large language models (LLMs) have evolved to be compressed versions of the world's knowledge. For instance, SimpleQA (Wei et al., 2024) benchmarks models based on whether they know in whose honor the 1877 Leipzig chess tournament was organized (it was Adolf Anderssen). But Morris et al. (2025) and Allen-Zhu & Li (2024) recently found that an LLM's storage is limited as a function of its number of parameters. Beyond a certain capacity threshold, exact factual storage becomes impossible, hence LLMs compress knowledge into lossy statistical predictions over tokens. While acceptable for some tokens, this inevitably introduces factual errors for others. (György et al., 2025).

This is particularly important for Small Language Models (SLMs, Belcak et al., 2025). An SLM has a strongly limited parameter count, often around or below 1B, and is thus neither capable of learning facts nor is meant to. Their goal is to quickly and cheaply predict tokens that they can (e.g., from context, or general language) and rely on tools and knowledge databases when they face factual predictions beyond their capacity. This is implemented by predicting some form of a `<CALL>` placeholder for the next token. This evokes two key research questions: (1) how can we keep the model's capacity free from trying to learn unlearnable tokens and instead call for help when necessary, and (2) what to do after we called for help? In this paper, we focus fully on the first question. For simplicity, for the second question we assume a bigger model to step in when the SLM calls, forming a model cascade (Varshney & Baral, 2022; Gupta et al., 2024).

From a learnability theory standpoint, the loss on the true token during pretraining indicates both whether the SLM predicts the true token correctly and whether it will reliably predict it after training. Loss-based

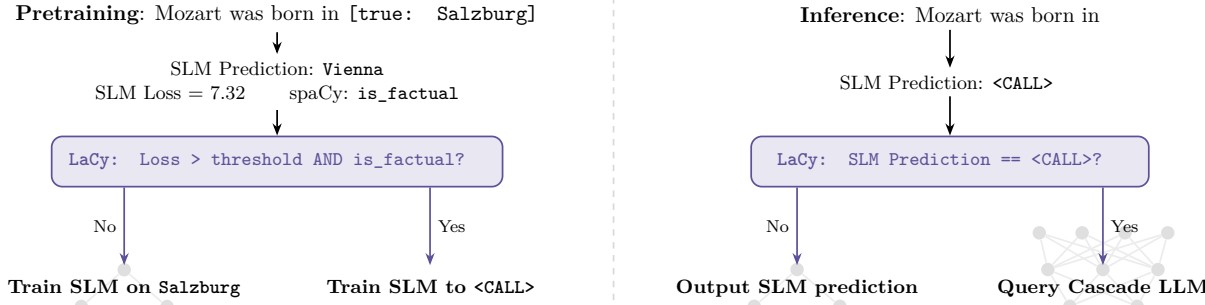

Figure 1: **Overview of the LaCy framework.** We decide which tokens an SLM can and should learn during pretrained based on its loss signal and a spaCy grammar processor. If it is a fact token that is too hard for this small model, we train to output a `<CALL>` token. At inference time, this triggers a larger model to step in. This enables the SLM to learn what it can predict, mitigating factual errors.

approaches like Rho-loss and Rho-1 (Mindermann et al., 2022; Lin et al., 2024) use this to decide which tokens to pretrain into the SLM's parameters and which to skip (or in our setting, to instead learn an explicit `<CALL>` token). However, an important finding that serves as the foundation of our work is that the cross-entropy loss is blind to the type of error: while the loss is indeed high for factual tokens that the SLM is not capable or meant to learn, it is also high when a predicted token just does not exactly match the ground-truth token because there are multiple acceptable continuations. For example, "The *cat*" and "The *time*" are both valid continuations of "The".

Rather than strict ground-truth matching via the loss, we argue that delegation decisions should depend on whether a predicted token would render the continuation factually or semantically invalid. We refer to this notion as *acceptability*. We find that in Wikipedia data, where factual information largely appears in isolated units, a spaCy-based grammar parser can detect factual tokens with only one truthful continuation (names, dates, etc.). If the SLM has a high loss on these tokens, it should `<CALL>`. Tokens with many acceptable continuations are less likely to cause factual errors, hence they are worth to learn even if their loss is high. Based on these insights, we propose LaCy: a novel pre-training method for Wikipedia-like domains, that augments SLM pretraining with this combination of grammar parsing and loss to decide which tokens to train on and for which to learn a `<CALL>` token instead.

We find that this simple and inexpensive change consistently improves the learning signal during pretraining. After training, the SLM has learned when to predict a `<CALL>` token and when to rely on its parametric knowledge to continue text. In downstream evaluations where we let the SLM write biographies in a model cascade with a larger model, this leads to higher FactScores (Min et al., 2023) than SLMs trained to predict `<CALL>` tokens based on losses (Wang et al., 2024a; Mindermann et al., 2022; Lin et al., 2024), LLM Judge annotations (Zhao et al., 2025), or token logits. These results support our central claim that loss alone is an insufficient signal for deciding what an SLM should learn versus delegate on Wikipedia-like domains. We discuss the domain-dependence and generality of our findings in detail in Section E.

## 2 Background

### 2.1 Why not learn all tokens?

The predominant mantra of foundation model training is to train models on as many tokens as possible. However, this strategy has come into question with the increasing understanding of how and what LLMs learn, and where they fail. The key insight is that language models can memorize facts only up to a limit dictated by their parameter size (Morris et al., 2025; Allen-Zhu & Li, 2024). After this, models seem to start "grokking", that is, they transition from nearly lossless to lossy predictions by overwriting and compressing parametric knowledge (Ghosal et al., 2025). While this appears useful for generalization, it poses a danger to trustworthiness: as pretraining progresses, the model associates more and more contexts it has seen some

time during pretraining with a rough statistical prediction. György et al. (2025) argue that not all contexts should be answered with a statistical prediction – indeed, some contexts, like facts, require exact predictions to prevent hallucinations.

Mitigating the drawbacks of statistical learning is especially important for the increasingly popular small language models (SLMs, Belcak et al., 2025). On the one hand, they are strictly limited in their capacity, but on the other hand, they are deployed with access to function calling or web queries to answer exact queries (Schick et al., 2023). Put differently, not only *can* an SLM not learn all tokens, it also *should* not. Rather, it should learn those tokens that are learnable *and* learn to identify those that are not and call out for help.

## 2.2 Which tokens are learnable?

The question of which tokens should be learned has recently received fresh attention. Generally, methods in this field pursue a selection mechanism and replace all non-learnable tokens with some instantiation of a `<CALL>` placeholder token. Zhao et al. (2025) replace all tokens with `<CALL>` tokens that GPT-4o (and a derived classifier) flags as factual knowledge. Other works propose mechanisms that are more adaptive to the model that is being trained. Chuang et al. (2025) propose to analyze which tokens a trained model is wrong on and then retrain the model without them. Cohen et al. (2024) similarly proposes to shift logits onto an `<IDK>` token if a predicted token is wrong.

A second category of methods rank which tokens do not appear learnable based on the difference of the loss of the SLM under training and of a reference model, which has seen more or higher quality data. They then disable gradient updates on a customizable portion of them, effectively ignoring those tokens in the backward step (Wang et al., 2024a; Lin et al., 2024). The idea is that the reference model's loss carries signal about how likely a token is to be wrong after more training. Instead of ignoring, an SLM can also use this signal to learn explicit `<CALL>` tokens. This is an instance of learnability theory. It has roots in domain adaptation (Moore & Lewis, 2010; Xie et al., 2023), distributionally robust optimization (Oren et al., 2019), and has recently resurfaced in Bayesian active learning Mindermann et al. (2022) and pretraining efficiency (Lin et al., 2024; Brandfonbrener et al., 2025). Our work refines these loss-based approaches by considering the token type.

Related work has also pinpointed limitations of the per-token cross-entropy loss as a measure of (factual) correctness. Mixed cross-entropy loss improves learning when multiple plausible outputs exist, using these outputs as alternative targets (Li & Lu, 2021). Concept-Aware LLMs instead compress text into categories such as Napoleon → French Emperor, which helps models learn conceptually important information, but may collapse truth-critical distinctions (Shani et al., 2023). We instead define acceptability as the key signal for learnability, and our delegation targets positions where the correct continuation is effectively unique, so there is a higher chance of an unacceptable mistake.

Our philosophy of *when to call* is also related to speculative decoding (Yan et al., 2025), whose rejection-style correction mechanism can be interpreted as a strictly inference-time operationalization of a model-relative, probabilistic notion of acceptability, where the large model's distribution defines the reference. In contrast, LaCy aims to capture a more objective notion of acceptability, grounded in truth and semantic validity rather than the beliefs of any particular model.

## 2.3 What to do once an SLM calls for help?

While implementing a lookup mechanism that is triggered after a `<CALL>` token is generated is not within the scope of our work—we focus exclusively on the question of when to call—various approaches have been considered to handle `<CALL>` tokens. The simplest option is to refrain from answering the query upon encountering a call (Cohen et al., 2024; Zhang et al., 2025; Chuang et al., 2025). On the other extreme, a call may trigger a database lookup Zhao et al. (2025) or a function call (Schick et al., 2023; Komeili et al., 2021). This may be the most forward-looking perspective on handling unlearnable contexts, but the mechanism of *when* to call becomes entangled with these specialized implementations of the lookup. Hence, in this paper, where we focus on when to call, we rely on a more generic way of handling calls: model cascades delegate the token to a more capable, but also increasingly costly, model (Varshney & Baral, 2022; Narasimhan et al., 2022; Jitkrittum et al., 2023; Gupta et al., 2024; Chen et al., 2024; Yue et al., 2024; Ding et al., 2024). This

gives an adaptive and well-performing plug-in for experiments in which we want to measure downstream improvements in factuality.

## 3 Loss alone cannot identify factual errors

The question of which tokens an SLM should learn depends on the downstream task. A universal goal is to avoid factual errors and hallucinations. Standard training minimizes the cross-entropy loss, which measures the model's likelihood of outputting the *exact continuation* that happens to be in the training document. We argue that the loss is not fully aligned with factual correctness, because some contexts can be continued in multiple valid ways, while others require very specific continuations. This discrepancy becomes important for token-selection: out of a limited budget of tokens we make models <CALL> on, it is crucial to choose those that are most likely to lead to factual errors. In order to measure how likely a token is to lead to factual errors, we propose the concept of *acceptability* as a relaxation of accuracy (whether a model's proposed next-token matches the ground truth in the data).

**Defining *Acceptability*.** Given a context, a proposed next-token is *acceptable* if, combined with the context, it produces a statement that is factually and logically consistent with the ground truth continuation and preserves its meaning. Although continuations that alter meaning may still be factually correct, we deem them unacceptable, as our goal includes training models to stick to the original data format and domain.

**Examples.** An *acceptable* continuation is: "Entre Campos Station is part of the *Lisbon*", if the original document is "Entre Campos Station is part of the *metro system in Lisbon*". An *unacceptable* continuation is: "Alan Turing was an English *linguist*" instead of "Alan Turing was an English *mathematician*".

**Experiment.** To test which token types are most prone to errors, we measure acceptability in a small-scale experiment: we pick 336 documents, covering ∼ 100k tokens out of the validation set of dwiki (a wikipedia dataset, Zhao et al., 2025), and score a 1.3B model's logits after training on 50B tokens. At each token position of a given document, we prompt Gemini 2.0 Flash (Google DeepMind, 2024) to assess acceptability of the model's proposed next token, given the true context and the ground truth next token. The details of the prompting are in Section A.

We qualitatively find two trends: non-acceptable tokens are usually predicted at positions where the ground-truth next token has high loss *and* is factual. To measure these trends quantitatively, we annotate the documents using a grammar parser, spaCy's small English web model (`en_core_web_sm`, Honnibal et al., 2020) for Named Entity Recognition and linguistic annotation, augmenting it with custom heuristics such as searching for common keywords and occurrence tracking. The details of the fact annotation can be found in Section A.2.1.

Figure 2 *(right)* confirms both observations: factual tokens and high-loss tokens have, on average, lower acceptability scores than their non-factual and low-loss counterparts. We repeat this experiment after training on only 10B tokens, and find similar behaviour (see Figure 9 in the Appendix).

This effect is invisible when only considering accuracy (predicted token equals ground truth)

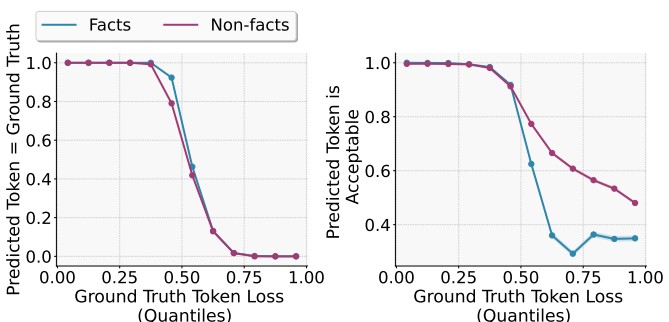

Figure 2: **The difference between Accuracy and Acceptability.** Token loss is predictive of whether a token is likely to match its exact ground-truth token *(left)*. However, this signal is blind to the type of token: Non-factual tokens are considered equally wrong as factual tokens, although non-factual tokens with high loss often do not render an output false *(right)*. We utilize a SpaCy grammar parser during pretraining to tell these two signals apart. Standard Errors are shown.

or loss (Figure 2 *(left)*). Our findings reveal that data selection based on only one of these signals is suboptimal: only considering loss or accuracy includes

acceptable non-factual tokens at the cost of missing inacceptable factual tokens, whereas simply selecting (a percentage of) factual tokens creates calls at factual positions that would have been acceptably answered.

Based on these findings, we propose a novel pretraining method, LaCy, that combines the loss signal with spaCy annotations to delegate via `<CALL>` tokens. Training with LaCy and retrieving next-tokens from a larger model at inference time whenever the SLM outputs a `<CALL>` token results in highly factual texts (see Figure 4). We verify this increase in factuality quantitatively in Section 5.4.

**What LaCy demonstrates.**   Importantly, LaCy is not meant to show that spaCy-based factuality is a universal delegation signal in any domain. We use it as a controlled operationalization of the broader claim established in this section: *token loss alone is insufficient to characterize which predictions an SLM can and should learn.* On Wikipedia, we found that the gap between accuracy/loss and acceptability is substantial, and that factuality information helps approximate this gap: high-loss factual tokens are much more likely to correspond to unacceptable continuations than high-loss non-factual tokens. We therefore use Wikipedia and lightweight spaCy annotations as a controlled testbed to show that domain-specific information (in this case, factuality) can improve delegation beyond loss alone by better approximating acceptability.

This also clarifies the scope of our claim. We view acceptability as a general target for deciding what a model should learn versus delegate, but effective approximations to acceptability may depend on the domain. While LaCy-style factual entity structures are a useful signal in Wikipedia-style generation, other domains may require different auxiliary signals. See more discussion on domain transfer in Section E.

## 4 LaCy: don't learn what you can't

In this section, we formalize the intuition of refining loss signals with spaCy parsing (LaCy) for SLM pretraining. Let $x = (x_1, x_2, \ldots, x_N)$ denote a data sequence, where each token $x_i$ is drawn from a fixed token dictionary $\mathcal{V}$. Autoregressive language models approximate the data distribution by next-token prediction, by fitting a distribution $p(x_{i+1} \mid x_{1:i}; \theta)$, parametrized by $\theta$.

LaCy modifies the standard negative log-likelihood objective by replacing ground-truth targets in each training batch with `<CALL>` tokens. LaCy's token selection combines spaCy-based factuality with loss-signals. Let $C_{\text{spaCy}} : \mathcal{V} \to \{0, 1\}$ be our custom function that flags factual tokens according to Section A.2.1. Our spaCy annotation alone assigns a fact label to 25% of tokens. However, based on Section 3, we do not want to delegate on all factual tokens, since there are some that may be predictable even for an SLM. We thus incorporate the loss signal, delegating the factual tokens with the highest loss. In Section 5.4, we ablate this design choice. With $x_i$ being the $i^{\text{th}}$ token in a mini-batch $\mathcal{B}$, we define the LaCy call mask as:

$$C_{\text{LaCy}}(x_i) = C_{\text{spaCy}}(x_i) \cdot \mathbb{I}\Big[i \text{ is in the top } n\% \text{ of } \mathcal{L}(\mathcal{B}; \theta)\Big],$$

where 1 denotes those tokens that LaCy changes to the `<CALL>` token. The modified pretraining objective is:

$$\mathcal{L}_{\text{LaCy}}(x; \theta) = -\frac{1}{N} \sum_{i=1}^{N} \Big[ C_{\text{LaCy}}(x_{i+1}) \log p(\texttt{<CALL>} \mid x_{1:i}; \theta)$$
$$+ (1 - C_{\text{LaCy}}(x_{i+1})) \log p_{\backslash \texttt{<CALL>}}(x_{i+1} \mid x_{1:i}; \theta) \Big],$$

where $p_{\backslash \texttt{<CALL>}}$ is the predictive token distribution excluding the `<CALL>` token, renormalized to probability 1.

To allow fair comparison to LLM judge-based factual annotations (Zhao et al., 2025), who delegate 15% of overall tokens (see Section A.2.2), we pick $n$ such that 15% of tokens are calls in each mini-batch. This means that the 60% highest-loss fact tokens are delegated and the 40% lowest-loss fact tokens, as well as all non-fact tokens, are learned as normal.

At inference time, LaCy generates text autoregressively until a `<CALL>` token is generated. The call is executed by prompting a larger cascade model with the context so-far (excluding the `<CALL>` token), and the output is appended to the generations, allowing the base model to continue. Further details can be found in Section A.

# 5 Experiments

We evaluate LaCy on factual precision, factual benchmarks, NLU, and validation losses against other `<CALL>` methods. In the main paper, we focus on 334M parameter SLMs. We also experimented with 1.3B models in Section F, reaching similar conclusions.

## 5.1 Experimental Setup

**Data.** We use the dwiki dataset, which consists of 3B tokens from the OLMo2 project (Groeneveld et al., 2024), as used by Zhao et al. (2025). We label the dataset using our strategy outlined in Section A.2.1, relying on the spaCy grammar parser (Honnibal et al., 2020). To compare to LLM judge annotations in our `<CALL>` delegation setup, we process the annotations of Zhao et al. (2025) as described in Section A.2.2.

**Pretraining.** We pretrain GPT-2 architectures from scratch with the SentencePiece tokenizer (Kudo & Richardson, 2018). The standard token dictionary of size 32,000 is extended by the special `<CALL>` token. Models are trained for $340 - 440$k iterations ($40-50$B tokens for 334M models and $35-45$B for 1.3B ablations, $\sim 16$ epochs), in accordance with literature recommendations for scarce data (Muennighoff et al., 2023). For our main experiments, the exact number of iterations was picked, similarly to past work (Lin et al., 2024) to equalize the number of tokens on which models receive gradient signals to the ground truth next token. This design choice equalizes the amount of information internalized by the model, allowing an apples-to-apples comparison of factual content. We note that this does not affect our main experiments (where all non-baseline methods are evaluated after equal 400k steps), and only impacts conclusions drawn from our ablations, which we detail in Section 5.4. This gives a token-to-parameter ratio of at least $120\times$ for 334M models and $25\times$ for 1.3B models. We use a context length of 1024 tokens and full precision. Details are in Section A.4.

**Inference and Cascading** We use greedy decoding. The cascade model used when an SLM defers is Llama 3.2 1B (Meta AI, 2024). Although this is a relatively small model, we found it to be particularly high-performing on wiki data, making it a good cascade partner. Whenever the SLM generates a `<CALL>` token, we pass the sequence generated so-far (including the prompt and excluding the `<CALL>` token) to the cascade model. In generation tasks, models are evaluated on an equal call budget of 22% of tokens, which we enforce by thresholding the `<CALL>` logit based on a running quantile that that adjusts based on the number of `<CALL>`s generated so far (details are in Section A.5). Since Llama does not use the SentencePiece tokenizer, occasionally it returns what in the SentencePiece tokenizer are multiple tokens, predominantly when retrieving 3-4 digit numeric tokens.

**Model comparisons.** We compare LaCy to a range of recent methods. For fairness, we reimplement and pretrain these methods with the same budgets and data.
- **Baseline**: pretrained without `<CALL>` delegations, evaluated at 340k steps to make up for the 15% of tokens that other methods do not train on.
- **Loss-based calls**: pretrained with uniformly sampled `<CALL>` masks. The SLM learns a constant prior logit on the `<CALL>` token, independent of context, and hence at inference time calls whenever the logits of all other tokens fall below this threshold, similar to Jitkrittum et al. (2023).
- **LLM judge**: pretrained with `<CALL>` delegations given by LLM judge annotations (Zhao et al., 2025).
- **Rho-1**: pretrained with `<CALL>` delegations chosen with low Rho-score (Lin et al., 2024). The original paper trains on tokens with high Rho score and skips the rest. We adapt this to our cascade setup by training on tokens with high Rho score and *delegating* on tokens with low scores.
- **LaCy**: pretrained with `<CALL>` delegations based on both spaCy factual annotations and loss signal.

## 5.2 Better Factual Accuracy in Biography Generation

We evaluate factual accuracy on a task that falls inside the learned distribution of our SLM. We prompt our models to generate Wikipedia-like biographies with their cascade partner.

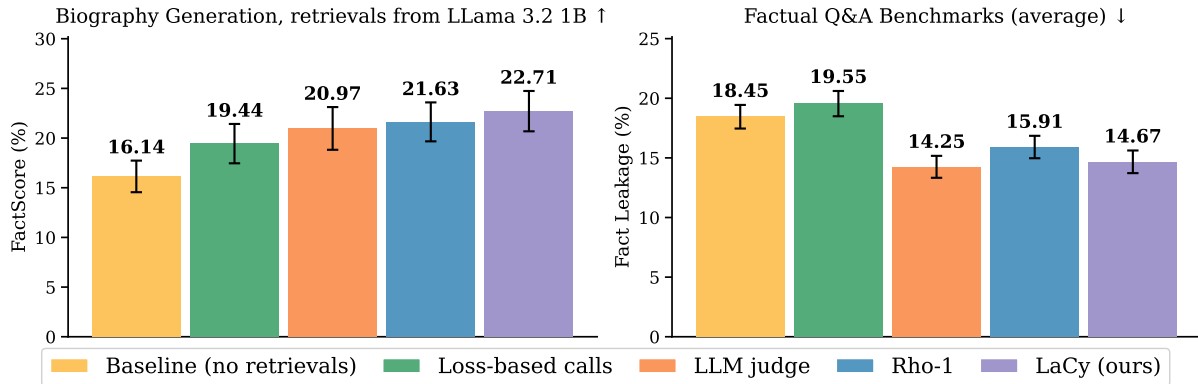

Figure 3: **Results overview for pretraining a 334M SLM.** *(Left.)* The LaCy-trained SLM achieves the highest FactScore when generating biography with Llama 3.2 1B as cascade partner, confirming that it successfully generates calls at factual token positions. *(Right.) Without calling*, LaCy has lowest fact leakage, meaning the least facts were trained into the limited parametric SLM memory. Error bars show 95% bootstrap confidence intervals (10,000 resamples).

*Tell me a bio of Errol Flynn. Errol Flynn is* **Baseline:** the name given to an American film starring Errol Flynn,directed by John Ford and produced by William Bowers. The screenplay was written by Robert L. Lippert based on his 1957 novel "The Man in the Iron Mask". It tells the story of a man who becomes involved with a woman he meets at a nightclub. The film was released theatrically in the United States on June 24, 1960, by Republic Pictures. Plot. In New York City, wealthy businessman Errol Flynn (Ferns) has been having an affair with his secretary, Jeanette (Bonnie Barnes). He is also secretly married to her, but she is not interested in him because he does not want to be seen as a husband for her.

*Tell me a bio of Errol Flynn. Errol Flynn is* **Rho-1:** an American actor, director and producer. He has been nominated for 4 Academy Awards (winning one in the category Best Actor for his performance as Captain James Bond) and won two Tonys. In addition to acting, he also directed "The Adventures of Tintin", " The Lion King" and "A Star Is Born". He was born on October 10, 29, 1876 in New York City; died January 3, 195 9 in Los Angeles from cancer at age ninety-three. Early life. Er rol Flynn grew up with three brothers and a sister in Brooklyn's Flatbush section. His father worked as a tailor while his mother ran a dress shop.

*Tell me a bio of Errol Flynn. Errol Flynn is* **LLM judge calls:** an Australian actor, director and producer who has appeared in more than 50 films since the early days of his career. He was born in 1897 in Sydney to parents from Ireland; he died on 14 August 1959 at age 62. Errol Flynn's first film was The Adventures of Tom Jones ( 193 2), which starred him as "Teddy". His other notable roles include Captain Blood in "The Wizard of Oz" (1, 19, 19) with Clive Owen, and the title role in "Sonny Boy", starring Mickey Rooney. In addition to acting, Flynn also produced and directed several short films including " A Christmas Carol "( 193 6).In December 193 6 it became known that Flynn had been approached by the British Film Institute for their upcoming feature film " Captain Blood."

*Tell me a bio of Errol Flynn. Errol Flynn is* **LaCy:** an Australian television personality, actor and film producer who has appeared in more than 100 films since the 193 0s as well as numerous TV series including " The Adventures of Robin Hood" (as Robin ), "The Adventures of the Black -clad Man", "Above & Beyond". He was also known for his role on the children's program " The Adventures of Robin Hood." Early life. Born in 1897 at home to parents John and Mary Flynn he grew up with his brother John Jr. His father died when he was young leaving him motherless. In 1905 he moved from rural Victoria to live nearer family where he attended school until moving back into the city after finishing highschool. Career. Acting career. He began acting professionally aged 16.

Figure 4: **Generations from 334 million parameter models.** The task is bibliography generation, the prompt is given in *italic*. `<CALL>` retrieved tokens from Llama 3.2 1B are highlighted in gray . Factual statements are colored in **green** for true, and **red** for false statements, as scored by FactScore (Min et al., 2023). LaCy and LLM judge call successfully delegate factual tokens, acquiring information on nationality, profession and dates. Rho-1 retrieves many useless tokens and has to rely on its own factual knowledge.

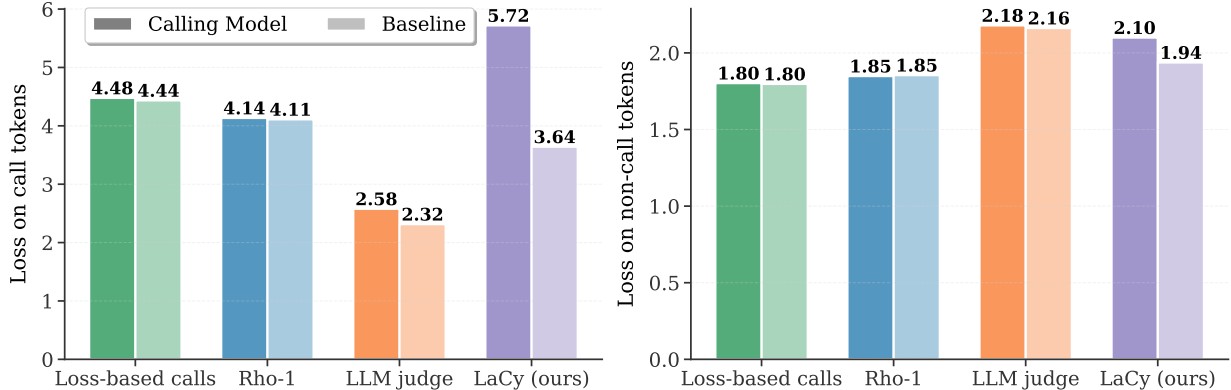

Figure 5: **Comparison of validation losses: LaCy distinguishes most the tokens it learns from the tokens it does not learn.** *(Left.)* Call losses. *(Right).* Non-call losses. For each `<CALL>`-augmented method, we construct its call mask by selecting the top 15% call logits in a batch. Full colors show the loss values of the `<CALL>`-augmented methods, while light colors show the loss of a vanilla baseline evaluated on the *same* `<CALL>` mask. LaCy calls on high-loss tokens (baseline call loss is high), and learns even less about them, achieving a call loss of 5.72. Its non-call loss is competitive with the factuality-based LLM judge.

**FactScore Results.** Factual accuracy is measured by FactScore (Min et al., 2023). FactScore breaks the generated biographies down into atomic facts and measures which proportion of generated facts is supported by the true Wikipedia page. Results are in Figure 3 (left). LaCy outperforms all previous methods, providing an increase of 6.88% compared to the baseline with no `<CALL>` augmentation. We use 10k bootstrap samples to compute 95% confidence intervals. We report symmetric uncertainties using the larger one of the upper and lower deviations. LaCy's confidence intervals are well separated from the baseline across all settings. For comparisons with LLM judge and Rho-1, individual intervals overlap due to the small evaluation set (183 entities), but the ordering consistently favours LaCy across both model sizes and all metrics. In Section F.4, we further interpret these results by comparison to off-the-shelf baselines. We find that 334M LaCy beats the 20× larger Llama 2-7B (Team, 2023).

LaCy's strength lies in querying in the right time: as illustrated by a sample generation (Figure 4), LaCy indeed learns to delegate when the next token is factual. Observe that not all facts inserted by the cascade partner are true (Llama 3.2 1B achieves 34.2% FactScore alone). This indicates LaCy's potential to perform even better with more factually accurate cascade partners. In the few exceptions of non-factual retrievals, such as where the retrieved token is `the`, the context suggests the possibility of a factual continuation. LLM judge qualitatively shows similar behavior, but has a slightly lower overall FactScore and a more complex training setup.

**RAG-Enhanced Cascade.** To increase the correctness of retrieved factual content, we ablate our cascade setup by using Qwen 3 32B (Qwen Team, 2025) as cascade partner, enhanced with a RAG prompt. The details of this setup can be found in Section A.5. The results (in Section F.3) stay consistent with our findings with Llama 3.2 1B cascade partner.

We emphasize that the focus of our work is the fundamental question of which tokens can and should be learned with an SLM. Our cascade setup is designed to provide a controlled experimental framework for comparing token delegation methods, and therefore it is intentionally simplified.

### 5.3 Decreased Fact Leakage

**Factual QA Results.** To analyze whether our SLM indeed does *not* internalize factual knowledge into its parametric memory (but instead call), we use a second evaluation based on QA datasets. We turn off calling capabilities by setting the `<CALL>` logit to $-\infty$ and prompt the models with questions (and sentence starts) on BigBench QA Wikidata (Srivastava et al., 2022) and the long-tail subset of PopQA (Asai et al.,

2024). We then check if the gold answer is contained in the generated answer. *Less* contained answers are better in this experiment. The reason we do not only measure FactScore to assess fact leakage is that FactScore generates long texts and the intervention on the `<CALL>` logit could drive subsequent generated tokens out-of-distribution, whereas on QA datasets we can prompt for isolated facts. Figure 3 (right) shows that LaCy achieves the lowest fact leakage, confirming its tendency to avoid learning facts.

**Validation Losses.** LaCy's low fact leakage is further supported by comparing validation losses on tokens where each method places calls ("call loss") versus does not ("non-call loss") in Figure 5. Note that Figure 5 should be interpreted with care because the methods choose different tokens to call on. Hence we provide the matched **Baseline** loss for each method, which is computed on the call or non-call mask proposed by each call-augmented model. LaCy achieves the largest validation loss of 5.72 on tokens it places calls on (Figure 5, left). Comparison to the baseline reveals that LaCy's calls happen on relatively high-loss, hence a-priori difficult tokens, and confirms the message: **the tokens LaCy chooses not to learn, it really does and should not learn.**

LaCy's validation loss on tokens it doesn't delegate but generates (Figure 5, right) is between non-call losses of solely loss-based methods (Loss-based calls, Rho-1) and the solely factuality-based LLM judge. This is explained by the insight that factuality is not always aligned with high loss: we have seen in Section 3 that some facts have low loss, hence their delegation increases the non-call loss.

### 5.4 LaCy ablations

In order to push the LaCy effect to its extreme, we explore the following ablations of LaCy (see also Section A.4.2):

- **spaCy only:** We remove loss-based thresholding from LaCy, and instead uniformly sample factual tokens to delegate, to create 15% calls per minibatch.
- **spaCy + Reference Model:** Similar to Rho-1, we use a reference model's loss instead of the SLM's own loss. This gives a signal on which tokens are "hard", independent of the SLM's training state.
- **LaCy + Ignorefacts:** We delegate facts using LaCy's selection, and additionally disable gradient updates on the remaining facts ($\sim 10\%$ of total tokens) in the spaCy annotation, effectively *ignoring* these tokens. To equalize the number of tokens models receive updates towards the true target, we allow training for 10% longer.
- **LaCy + Ignore:** We delegate facts using LaCy's selection, disable backpropagation on all remaining facts *and additionally* on some non-factual and non-grammatical tokens (defined in Section A.2.1) with the highest loss. This totals to delegating 15% and ignoring 15% of tokens per minibatch. We allow for training for 15% longer.

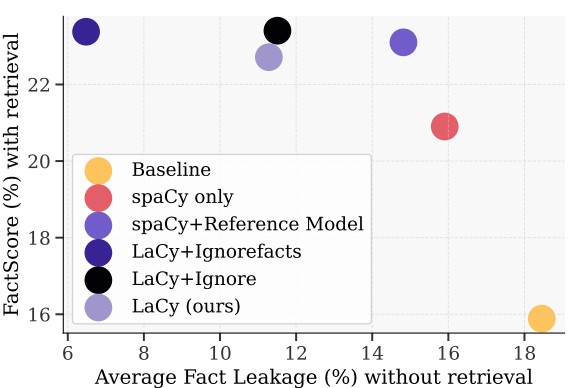

Figure 6: **FactScore (with cascade) against fact leakage (without cascade) for LaCy ablations.** Methods disabling backpropagation on $x\%$ tokens are evaluated after $x\%$ more training steps. Loss signal is beneficial: spaCy (without loss) performs worse. Using a reference loss or ignoring non-delegated facts gives marginal improvements on FactScore at a computational overhead (Table 2). Offloading even more tokens is beneficial.

Figure 6 shows that the loss-based selection component of LaCy is necessary: spaCy only performs worse both on FactScore and Fact Leakage. Section F.5.2 in the Appendix further disentangles the relative contributions of the spaCy and loss components and related methods, via analyzing the overlaps of delegated tokens. At 15% delegation budget, loss-only selection has 62.9%, while spaCy-based factuality has 54.6% overlap with the LaCy delegated tokens. This shows that both components are integral to LaCy's success, with the loss playing a slightly larger part. Switching the loss to a reference model's loss (spaCy + Reference Model) gives very minor benefits to FactScore, likely because the reference loss provides more consistent signal on which

tokens have high loss. However, this is at the cost of an overhead similar to Rho-1 (Table 2), and requires two-stage training where first a complete model is trained, only to then restart training a new model. Ignoring the remaining facts (LaCy + Ignorefacts) or even more tokens (LaCy + Ignore) is slightly beneficial, but only if the number of backpropagated tokens are equalized. Figure 14 in the Appendix shows that improvements of ignoring tokens on FactScore disappear once methods are evaluated on an equal number of forward steps. We suspect that there are conflicting forces at play: while not learning any facts creates a more consistent `<CALL>` signal, as Figure 2 in Section 3 has shown, some facts *can* be learned. Overall, LaCy provides the best tradeoff between factual accuracy and pre-training budget.

**The impact of *how much* LaCy calls.** In Section F.5.4, we replicate our experiments varying the call ratio hyperparameter both at training and inference-time. At a train-time calling ratio of 10%, we see similar trends to the main results: LaCy (ours) consistently outperforms other methods on FactScore, and almost always on fact leakage. Nevertheless, factual accuracies are slightly lower than at 15% delegation, our original configuration. At inference time, more delegation is helpful, but only marginally: at +60% delegation ratio, we see around 2% increase for 334m and 6% for 1.3B, showing that our initial 22% call ratio provides a good trade-off between performance and delegation ratio.

**The impact of in *which training stage* LaCy calls.** In Section F.5.5, we study the effect of delegation with LaCy at *pretraining time* as opposed to *only at inference-time*. We implement an inference-time only version of LaCy, that delegates whenever the model's next-token prediction is a fact (as annotated by spaCy on the fly), and the model's "confidence" in the response, measured by the log probability of the top next-token, is low. Results show the benefit of pretraining-time delegation.

## 5.5 Not learning facts does not worsen NLU

Factual knowledge and Natural Language Understanding (NLU) are considered separate skills. We test this hypothesis by evaluating our `<CALL>` models without cascading on SLM-appropriate NLU benchmarks. Table 1 confirms, in accordance with previous work (Zhao et al., 2025) that fact offloading neither increases, nor decreases NLU ability significantly. This means that factual knowledge is not needed for NLU tasks, but, interestingly, freeing model capacity by

Table 1: **NLU performance of `<CALL>` augmented models *without* cascade.** We confirm that factual offloading does not significantly degrade Natural Language Understanding (NLU).

| Model | Metrics | | | | |
|---|---|---|---|---|---|
| | ARC Easy | HellaSwag | PIQA | SIQA | Average |
| Random chance | 25.0 | 25.0 | 50.0 | 33.3 | 33.3 |
| Baseline | 34.8 | **28.8** | 59.0 | 35.9 | 39.6 |
| Loss-based calling | 34.3 | 28.6 | 57.1 | 36.3 | 39.1 |
| Rho-1 | 35.0 | 28.6 | 56.8 | 35.9 | 39.1 |
| LLM judge | 33.8 | 28.3 | 57.3 | **36.8** | 39.1 |
| LaCy | **35.6** | 28.5 | **59.3** | 36.2 | **39.9** |

offloading facts does not improve NLU. Table 11 in the Appendix shows that offloading more than factual tokens degrades NLU performance.

## 5.6 Throughput overhead is minimal

Both LaCy and the methods we compare against require a certain labeling effort before training (except loss-based, since loss is computed anyways during training). In Table 2 we report the overhead that this causes. LLM judge, which uses a large additional LLM, implies most cost to iterate over the pretraining dataset. The spaCy labeling that LaCy uses runs on CPU cores. Not only does this scale cheaper than GPUs, but it can also be

Table 2: **Overhead of producing pretraining labels.** LaCy is the only method (except the loss-based baseline) whose labeling does not require a GPU, allowing to scale to large pretraining datasets.

| Method | Preprocessing Overhead | Device |
|---|---|---|
| Loss-based | None | None |
| LLM judge | 233 h/1B tokens | single A100 GPU |
| Rho-1 | 56 h/1B tokens | single A100 GPU |
| LaCy | 152 h/1B tokens | single CPU core |

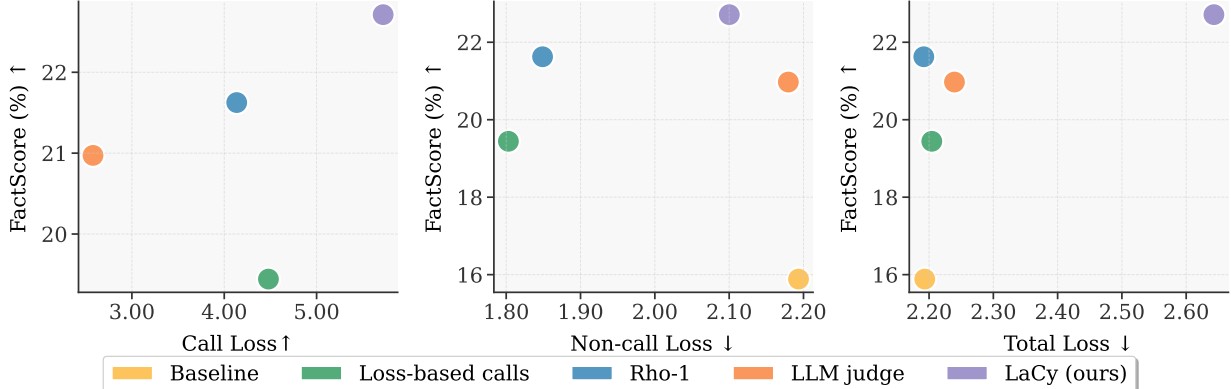

Figure 7: **Validation loss is not correlated with FactScore.** Neither the call loss *(Right)*, non-call loss *(Middle)*, nor the total loss *(Left)* is predictive of the FactScore of the displayed methods. Findings linking loss with downstream performance in related work Kaplan et al. (2020); Srivastava et al. (2022); Krajewski et al. (2025) do not transfer to our token-selection setting.

included in the dataloader online during
training, without occupying GPU cycles. This makes LaCy compatible with larger-scale pretraining.

### 5.7 Loss is Not Correlated with FactScore

Models' abilities, especially during pretraining, are often assessed by their validation loss. Indeed, in most training scenarios, the validation loss correlates with the model's downstream performance and reasoning abilities Kaplan et al. (2020); Srivastava et al. (2022); Krajewski et al. (2025). Interestingly, we observe (Figure 7) that *in our setup*, none of the validation losses we consider (call loss, non-call loss, total loss) are aligned with factual accuracy (measured by FactScore). This is because token-selection and delegation implicitly changes the models' target distribution (i.e., which tokens we evaluate on), hence methods are no longer comparable based on losses alone, even when compared to a baseline (in Figure 5). We thus recommend to evaluate cascaded models in setups similar to ours on downstream tasks, like FactScore.

## 6 Conclusion

Our findings suggest that "which tokens to delegate" is a nuanced question in an SLM setting. Human heuristic notions may overcall on tokens that are predictable even for a small model, such as the word "Party" when already given the context "politician for the Moderate". On the other hand, fully automated notions based on the loss are blind to semantic issues: the model may achieve relatively small loss while predicting a token that is completely wrong and might have a high loss on a token where it placed probability mass on an acceptable synonym of a word. The method proposed in this work, LaCy, shows that incorporating these nuances can lead to effective and yet simple training for SLMs.

However, we note that this study is an explorative pilot study. The model sometimes tries to predict factual tokens it should not, which we believe is mostly because it was trained at a small scale. We are confident that larger-scale training will make the behaviors more robust, because they are consistent across experiments and follow what we expect theoretically. A second point to follow up on is the question we excluded from this paper: what to do once the SLM calls. A general limitation of cascade systems: mitigation of factual inaccuracies and hallucinations is achieved via allowing more capable models handle factual queries. While reducing inaccuracies, this approach still encounters the risk of hallucinations from cascade partners. Previous work has proposed exciting joint architectures both with cascade models and classical knowledge bases. We expect that our improvements on *when* to call will transfer to the overall performance of those systems.

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

# A    Implementation Details

## A.1    Assessing acceptability in Section 3

We assess the *acceptability* of predicted tokens using an LLM as a judge. We fix a validation batch of size 112. Starting from token 1, at each position of each tokenized document, we record the model's proposed next token (corresponding to the highest logit, i.e., greedy decoding) alongside the the ground truth next token.

We prompt Gemini 2.0 Flash to score the semantic and factual validity of the proposed next token using the prompt below. We manually verified that this matches our intended intuition of acceptability.

---

Prompt for Judging Acceptability.

```
# Task
You must evaluate whether a proposed next token is a valid continuation of a
    given text.

# Input
You will receive three pieces of information:
1. **starting_text**: The initial text segment
2. **proposed_next_token**: A token that could continue the starting_text
3. **reference_next_token**: A reference token for comparison

# Evaluation Criteria
The proposed_next_token is VALID if:
- Joining starting_text + proposed_next_token creates a statement that is
    logically and factually compatible with starting_text +
    reference_next_token
- The two resulting statements do not contradict each other
- The two resulting statements have similar meaning

The proposed_next_token is INVALID if:
- Joining starting_text + proposed_next_token creates a statement that
    contradicts, conflicts with, or significantly changes the meaning of
    starting_text + reference_next_token

# Output Format
Provide:
1. **explanation**: Your reasoning for the decision
2. **output**: A binary label (1 = valid, 0 = invalid)

# Examples

## Example 1: Invalid Continuation
**starting_text**: 'Wolfgang Amadeus Mozart\n\n (27 January 17'
**proposed_next_token**: '6'
**reference_next_token**: '5'
**explanation**: The proposed_next_token does not match reference_next_token.
    Completing with '6' would create 'Wolfgang Amadeus Mozart (27 January
    176...', while the reference creates '...175...'. Mozart was born in
    1756, and the reference token '5' indicates the correct continuation is
    1756 (starting with 175). The digit '6' creates a factual conflict
    because it would lead to an incorrect year.
**output**: 0

## Example 2: Invalid Continuation
**starting_text**: 'Alan Turing was an English '
```

```
**proposed_next_token**: 'linguist'
**reference_next_token**: 'mathematician'
**explanation**: The proposed_next_token does not match reference_next_token.
    Completing with 'linguist' would create 'Alan Turing was an English
    linguist', while the reference creates 'Alan Turing was an English
    mathematician'. This creates a significant difference in meaning. Hence
    the continuation is invalid.
**output**: 0

## Example 3: Valid Continuation
**starting_text**: 'Entre Campos (Lisbon Metro)\n\nEntre Campos station is
    part of the '
**proposed_next_token**: 'metro'
**reference_next_token**: 'Yellow'
**explanation**: The proposed_next_token does not match reference_next_token,
    but it does not create a factual conflict. The reference would create '
    Entre Campos station is part of the Yellow [line]', while the proposal
    creates 'Entre Campos station is part of the metro [network/system]'.
    Both statements have similar meaning and are factually true and
    compatible - the station IS part of the Yellow line AND part of the metro
     system. These are not contradictory facts.
**output**: 1

# Your Task
**starting_text**: '{}'
**proposed_next_token**: '{}'
**reference_next_token**: '{}'
**explanation**:
```

For each query, the outputted score is extracted and averaged across all model queries.

## A.2   Data Preparation

We use the dwiki dataset, which consists of (∼3B tokens) from the OLMo2 project (Groeneveld et al., 2024), previously used by Zhao et al. (2025).

### A.2.1   Data processing with spaCy

We implement an NLP-based token classification system that categorizes each token in a document into one of three semantic classes: *grammatical* (e.g. prepositions, punctuation), *factual* (first occurrences of informative content), and *other* (repeated or not factually-essential content). In the main paper, we then use the differentiation factual vs non-factual (grammatical and other). We use spaCy's small English web model (`en_core_web_sm`, Honnibal et al., 2020) for an initial linguistic labeling, and augment it with custom heuristics to improve entity recognition and occurrence tracking. We further customize fact annotation from word to token-level to make it suitable for autoregressive language model training.

Before detailing each step, there is one key ingredient when deciding when an SLM should delegate. As autoregressive language models, the first mentions of entities and concepts are hard-to-learn facts, and should be delegated due to the SLM's limited parametric knowledge capacity. When predicting the second mention of an entity, autoregressive models have access to the previous mention in the context, and hence parametric knowledge is not needed for predicting the second mention. Therefore, our approach differs from fact annotation: these words are factual per se, but for the purpose of training SLMs, factual knowledge is not needed for learning them, and so we do not label them as factual tokens.

Our pipeline annotates words in a document as facts in the following steps:

1. **Named Entity Processing.** We process spaCy's named entities and log their occurrence. Only the first mention of a named entity is classified as a fact. For `PERSON` type words, we check if any name component was seen before (for example, `Wolfgang` can be a second mention for `Wolfgang Amadeus Mozart`). For other named entity categories (such as `ORG` and `DATE`), only the full entity counts as a repetition.

2. **Supplementary Entity Detection (beyond spaCy's Named Entity Recognition).** First, we process spaCy's noun chunks as follows: noun chunks spanning whitespace boundaries (e.g., newlines) are split into separate sub-chunks to ensure accurate word boundary detection. Then, we process these chunks, and classify **their first occurrence** as a fact whenever

   - They are likely `PERSON` based on
     - syntactic role: subjects (`Marie Curie discovered radium.`) and appositives (`The physicist, Marie Curie, discovered radium.`) suggest person names;
     - contextual cues are present (following verbs like `born` or `died`, preceding titles like `Dr` or `Professor`)
   - They are likely an `ORG`: keywords such as `committee`, `council`, `university` appear in the noun sequence, or there is a leading definite article `the`
   - They are proper nouns (capitalized words denoting specific entities, e.g. `Mount Everest`)
   - They are common nouns, but are likely factual: words serving as predicative attributes (`She was a lawyer`), direct objects (`She studied physics`), or appositives (`Marie Curie, a physicist, discovered radium.`) are considered factual, while those governed by manner prepositions (`he was a lawyer by training`) are not.
   - They are numeric (likely `DATE`) words: we classify all first occurrences of numeric words as facts (where multiple-digit numbers like `1987` are treated as a single number)

3. **Classification of *grammatical* words.** We assign *grammatical* label to determiners (e.g. `an, this, my, each`), prepositions (e.g. `on, until`), conjunctions (e.g. `and, unless`), auxiliaries (e.g. `have, might`), and punctuation (e.g. `-, ?`).

4. **Classification of *other* words.** Words not classified so far are labeled *other*. For the purpose of some ablations (Table 5), we distinguish the *other* category from the *grammatical* category, but we merge them in the main paper.

After word-level annotation, we tokenize each document using the SentencePiece tokenizer (Kudo & Richardson, 2018) and assign classes to subword tokens based on the class of the source word they belong to. When a word is split into multiple subword tokens, all resulting tokens inherit the label of the original word.

The full annotation pipeline, including tokenization, requires 22.5 hours on 32 CPUs (the results in Table 2 reports slightly faster values because it reports pure throughput within the loop, without setup costs). We chose an offline annotation because we ran multiple training runs, but the above speed and simplicity of the method would also allow to run it online as a part of the dataloader when working on large-scale pretraining datasets.

**Alternatives to spaCy for factual annotation.** Beyond spaCy-based linguistic annotation, several alternative approaches could be used to identify factual tokens. One option is other Python NLP frameworks such as Stanza (Qi et al., 2020) or Flair (Akbik et al., 2019), which provide named entity recognition and syntactic analysis similar to spaCy albeit at slightly slower speed. A more expensive alternative is to use transformer-based encoder models (e.g., BERT (Devlin et al., 2019) or DeBERTa (He et al., 2021)) and fine-tune with a token classification head to predict whether a token is factual. Labels could be obtained from spaCy, LLM-judge annotations, or human annotation. Compared to spaCy or rule-based approaches, these models may detect richer contextual information, but require additional labeled data, training, and computational resources. At the most powerful end of the spectrum are LLM-judge based annotation methods, which can directly judge whether a token or span corresponds to factual knowledge, but incur substantially higher computational cost (Zhao et al., 2025), and are less reproducible. We chose spaCy because it provides

a simple, reproducible, and computationally efficient approximation of factual token positions that scales to large pretraining corpora, while avoiding the training requirements of supervised classifiers and the expense of LLM-based annotation. This fits with our research aim of using spaCy-based factuality as a tool to illustrate that token loss alone is insufficient to characterize which predictions an SLM can and should learn.

### A.2.2 LMLM data Processing

For fair comparison with LMLM's data selection driven by an LLM judge annotator (Zhao et al., 2025), we process their entity-level factual annotations (available at `kilian-group/LMLM-pretrain-dwiki6.1M`) to be compatible with our cascaded setup. This consists of removing database lookup calls, and turning each delegated word into a sequence of `<CALL>` tokens. This creates `<CALL>` and `<NONCALL>` labels for every token in the dataset. During training, the clean data is passed to the model, the `<CALL>` labels are only included as targets in the loss function. An example is given below.

Clean example snippet: `Napoleon was born on August 15, 1769.`

Snippet processed by Zhao et al. (2025): `Napoleon was born on <|db_start|> Napoleon <|sep|> Birth_Date <|db_retrieve|> August 15, 1769.` (where August 15, 1769, is filled in automatically after the lookup)

Processed example (used as target in the loss function): `Napoleon was born on <CALL> <CALL><CALL>, <CALL><CALL><CALL><CALL>.`

We note that this is not a critique of their labeling. We just convert to our format in this study to focus fully on when to call, rather than in which format to call.

### A.3 Model Architectures

We train GPT2 style transformers (Radford et al., 2019) of two different scales, 334 million and approximately 1.3 billion parameters. The architectures are described in Table 3. We fix the vocabulary size to 32,001 (including the special `<CALL>` token, and the sequence length to 1,024. We use the SentencePiece tokenizer (Kudo & Richardson, 2018). Computations are performed with precision bfloat16, apart from normalization layers and softmax in self-attention, which we compute with precision float32, following standard practices (Rabe & Staats, 2021; Wang et al., 2024b).

| Model Size | Parameters | Dim | Heads | Layers |
|---|---|---|---|---|
| Medium | 334m | 1024 | 16 | 24 |
| XL | 1.27B | 2048 | 16 | 24 |

Table 3: **Model configurations for different sizes of GPT models.** All models share the same tokenizer with vocabulary size of 32,001 and MLP dimension equaling 4 times the dimension of the model. We include the embedding layer in the parameter count.

### A.4 Pretraining

### A.4.1 Hyperparameters

We train most of our models on 8 A100-80GB GPUs, except for those requiring a reference model. For these, we compute the reference model loss online, sharding both reference and target models across 2 A100-80GB GPUs. Training on 8 GPUs finishes in 3 days. To allow for large batch sizes, we use gradient accumulation across 4 steps. We use AdamW with a weight decay of 0.1, with warmup only and no other learning rate scheduling. Hyperparameters such as learning rate, warmup steps and precision are detailed in Table 4. Calling models receive gradient signals on only 85% of tokens, which is why they are trained 15% longer than their corresponding baseline models. The only exception is the method "Loss + Ignorefacts", where we further compensate for the 10% of fact tokens that are neither learnt, nor delegated. We train these models

| Model | Batch Size | Total Steps | Learning Rate | Warmup | Precision |
|---|---|---|---|---|---|
| 334m calling models | 128 | 400k | 2e-4 | 2560 | FP32 |
| 334m baseline model | 128 | 340k | 2e-4 | 2560 | FP32 |
| 1.3B calling models | 112 | 400k | 1e-4 | 6400 | FP32 |
| 1.3B baseline model | 112 | 340k | 1e-4 | 6400 | FP32 |
| 1.3B (reference models) | 112 | 2540k | 1.5e-4 | 6400 | FP32 |

Table 4: **Training hyperparameters.**

for 440k steps. Reference models are trained on 2,540,000 steps, which includes 1,280,000 steps of initial pretraining on RedPajama V2 (Weber et al., 2024).

### A.4.2 Training Losses

Let $x = (x_1, x_2, \ldots, x_N)$ denote a data sequence, where each token $x_i$ is drawn from a fixed token dictionary $\mathcal{V}$. Autoregressive language models approximate the data distribution by next-token prediction, by fitting a distribution $p(x_{i+1} \mid x_{1:i}; \theta)$, parametrized by $\theta$, which typically denotes the parameters of a neural network, and is obtained by minimizing the negative log-likelihood:

$$\mathcal{L}(x; \theta) = -\frac{1}{N} \sum_{i=1}^{N} \log p(x_{i+1} \mid x_{1:i}; \theta). \tag{1}$$

In this work, we focus on token selection, and either delegate or ignore unselected tokens. This gives rise to modified objectives. Let $I, C$ is an arbitrary binary masks that define which tokens to ignore and call on, respectively.

**Training loss with Ignore Tokens.**

$$\mathcal{L}(x; \theta) = -\frac{1}{\sum_{i=1}^{N} (1 - I(x_i))} \sum_{i=1}^{N} (1 - I(x_{i+1})) \cdot \log p(x_{i+1} \mid x_{1:i}; \theta), \tag{2}$$

**Training loss with Call Tokens.**

$$\mathcal{L}(x; \theta) = -\frac{1}{N} \sum_{i=1}^{N} (1 - C(x_{i+1})) \cdot \log p_{\backslash \texttt{<CALL>}}(x_{i+1} \mid x_{1:i}; \theta) + C(x_{i+1}) \cdot \log p(\texttt{<CALL>} \mid x_{1:i}; \theta), \tag{3}$$

where $p_{\backslash \texttt{<CALL>}}$ denotes the operation of setting the `<CALL>` token's logit to $-\infty$. Combining the two losses, we obtain

**Training loss with Ignore and Call Tokens.**

$$\mathcal{L}(x; \theta) = -\frac{1}{N} \sum_{i=1}^{N} (1 - C(x_{i+1})) \cdot (1 - I(x_{i+1})) \cdot \log p_{\backslash \texttt{<CALL>}}(x_{i+1} \mid x_{1:i}; \theta) + C(x_{i+1}) \cdot \log p(\texttt{<CALL>} \mid x_{1:i}; \theta), \tag{4}$$

Notice that the context $x_{1:i}$ is unchanged, hence token-selection is only applied at backpropagation.

**Token-Selection Masks.** Let $L : \mathcal{V} \to \{0, 1\}$ be a labeling function that flags facts, and let $L_{\text{LLM judge}}$ and $L_{\text{spaCy}}$ denote the labeling functions corresponding to each data processing technique. For token $x_i$ within batch $\mathcal{B}$, our token-call masks and ignore masks are defined in Table 5.

### A.4.3 Validation Losses

We reserve a randomly chosen 10% of the dwiki dataset as validation set. For calling models, we decode 15% of predictions as `<CALL>`s as follows. We record the positions where the call logit is the top logit. If this occurs for more than 15% of tokens, we cap to 15%, keeping the positions with highest call logits. If the call logit is the top logit for less than 15% of tokens, we add the next highest call logits to reach 15% calls. This way, we extract a call mask $C_M$ from each model. We can then define

**Call loss:**

$$\mathcal{L}_{\text{Call}}(x; \theta, C_M) = -\frac{1}{\sum_{i=1}^N C_M(x_i)} \sum_{i=1}^N C_M(x_{i+1}) \cdot \log p(x_{i+1} \mid x_{1:i}; \theta) \tag{5}$$

**Non-call loss:**

$$\mathcal{L}_{\text{Non-call}}(x; \theta, C_M) = -\frac{1}{\sum_{i=1}^N (1 - C_M(x_i))} \sum_{i=1}^N (1 - C_M(x_{i+1})) \cdot \log p_{\backslash \texttt{<CALL>}}(x_{i+1} \mid x_{1:i}; \theta). \tag{6}$$

When comparing call and non-call losses of model $M$ to a baseline (such as in Figure 21), the baseline model's loss is computed using the call mask $C_M$.

| Method Name | Call Mask $C(x_i)$ | Ignore Mask $I(x_i)$ |
|---|---|---|
| Baseline | None | None |
| Loss-based calls | $C(x_i) = v$, where $v \sim \text{Bern}(0.15)$ | None |
| Rho-1 | $\mathbb{I}\big[i \text{ ranks in the bottom 15\% of } \mathcal{L}(\mathcal{B};\theta) - \mathcal{L}_{\text{RefModel}}(\mathcal{B};\theta)\big]$ | None |
| LLM judge | $L_{\text{LLM judge}}(x_i)$ | None |
| spaCy | $L_{\text{spaCy}}(x_i)$ | None |
| **LaCy** | $L_{\text{spaCy}}(x_i) \cdot \mathbb{I}\big[i \text{ ranks in the top } n\% \text{ of } \mathcal{L}(\mathcal{B};\theta)\big]$ | None |
| spaCy+Refloss | $L_{\text{spaCy}}(x_i) \cdot \mathbb{I}\big[i \text{ ranks in the top } n\% \text{ of } \mathcal{L}_{\textbf{RefModel}}(\mathcal{B};\theta)\big]$ | None |
| LaCy + Ignorefacts | $L_{\text{spaCy}}(x_i) \cdot \mathbb{I}\big[i \text{ ranks in the top } n\% \text{ of } \mathcal{L}(\mathcal{B};\theta)\big]$ | $L_{\text{spaCy}}(x_i) \cdot (1 - C(x_i))$ |
| LaCy + Ignore | $L_{\text{spaCy}}(x_i) \cdot \mathbb{I}\big[i \text{ ranks in the top } n\% \text{ of } \mathcal{L}(\mathcal{B};\theta)\big]$ | $L_{\text{spaCy}}(x_i) \cdot (1 - C(x_i)) +$ $\mathbb{I}\big[i \text{ ranks in top } k\% \text{ of } \mathcal{L}(\mathcal{B}_{\backslash\text{grammar}};\theta)\big]$ |

Table 5: **Definitions of call and ignore masks for all methods in our paper.** The call masks are always chosen such that 15% of tokens are calls in each batch. In LaCy + Ignore, $k$ is chosen such that there are 15% tokens ignored in each batch.

## A.5  Inference and Cascading

**Inference.**  We fix a maximum generation length of 256, and use greedy decoding with a repetition penalty 1.2.

**Cascade Model.**  For delegation, we use the off-the-shelf model Llama-3.2-1B, which we load with its own tokenizer.

**Cascade for Open-Ended Generation.**  The cascade is carried out as follows. Whenever a `<CALL>` token is generated, we pass the sequence generated so-far (excluding the `<CALL>` token) to the cascade model. We extract the highest-probability token and append to the generated text. The map from the cascade model's tokenizer to the base model's tokenizer is not bijective. If the retrieved token is not present in the base model's token dictionary, we choose the second highest-probability token of the cascade model. There are some cases when the retrieved token maps to more than one token in the base model's dictionary. In these cases, to avoid wasting the expertise of the cascade model, we append all of these tokens to the generated text. This happens in around 15% of retireval queries, and no more than three tokens get retrieved in a single query. The most notable example is the mismatch between the tokenization of numbers: Llama-3.2-1B encodes three-digit numbers as single tokens, whereas our SentencePiece tokenizer handles each digit separately (Kudo & Richardson, 2018).

**RAG Setup.**  To increase factual accuracy, we evaluate SLMs with cascade partner Qwen 3 32B (Qwen Team, 2025) enhanced with a RAG prompt given below. The background information `wiki_content` is obtained by using the full text from the wikipedia entry corresponding to each given person, truncated to 8000 characters (roughly 2000 tokens). For those few entities who do not have a unique wikipedia page, no background information was provided. We use Qwen 3 32B with greedy decoding and a repetition penalty of 1.2. The Qwen 3 32B + RAG setup, when evaluated on its own, achieves a FactScore of 79%.

---

**RAG Prompt for Qwen 3 32B.**

```
f"""<|im_start|>system
    You are an assistant who writes biographies of famous people and events.
        Continue any given text naturally and fluently.<|im_end|>
    <|im_start|>user
    Write a short biography about {person_name}. Here is some background
        information:
    {wiki_content}<|im_end|>    # No \n between the last sentence end and <|
        im_end|>
    <|im_start|>assistant
    <think>

    </think>

    {original_text}"""
```

---

**Calibration of the Calling Ratio.**  In order to assess models in equal conditions, we calibrate the `<CALL>` token's appearance rate by estimating a running threshold on the `<CALL>` token's logit during generation. Due to the nonstationarity of the call logits during generation, the 15% target calling ratio corresponds to an actual calling ratio of about 22% across all methods.

# B  Pseudocode

For reproducibility, we supply pseudocode implementations of train-time (Algorithm 1) and test-time delegations (Algorithm 2) in LaCy.

---

**Algorithm 1** LaCy train-time delegation

---

**Require:** Token minibatch $X$, factuality mask $F$, padding mask $A$, delegation ratio $r$, model $p_\theta$

    *Align the inputs, next-token targets, and annotation masks.*

1: $X_{\text{in}} \leftarrow X[:, :-1]$
2: $Y \leftarrow X[:, 1:], \quad F \leftarrow F[:, 1:], \quad A \leftarrow A[:, 1:]$

    *Compute losses on the original targets for ranking. Set CALL logit to $-\infty$ as calls are not an option yet.*

3: $Z \leftarrow p_\theta(X_{\text{in}})$
4: $Z^{\neg\text{CALL}} \leftarrow \text{MaskCallLogit}(Z)$
5: $\ell \leftarrow \text{CE}(Z^{\neg\text{CALL}}, Y, \text{reduction} = \text{none})$

    *Select the highest-loss factual tokens across the minibatch.*

6: $K \leftarrow \lfloor r \sum_i A_i \rfloor$
7: $C \leftarrow \text{TopKMask}(\ell, \ K, \ F \wedge A)$

    *Replace selected targets with CALL locally in this minibatch.*

8: $\widetilde{Y} \leftarrow Y$
9: $\widetilde{Y}[C] \leftarrow \langle\text{CALL}\rangle$

    *Train on CALL at delegated positions and on the original target elsewhere.*

10: **for all** non-padding positions $i$ **do**
11:     **if** $C_i = 1$ **then**
12:         $L_i \leftarrow \text{CE}(Z_i, \langle\text{CALL}\rangle)$
13:     **else**
14:         $L_i \leftarrow \text{CE}(Z_i^{\neg\text{CALL}}, Y_i)$
15:     **end if**
16: **end for**
17: $\mathcal{L} \leftarrow \dfrac{\sum_i A_i L_i}{\sum_i A_i}$
18: Update $\theta$ using $\mathcal{L}$

---

---

**Algorithm 2** Test-time delegation

---

**Require:** Prompt $X$, small model $p_\theta$, cascade partner $q$, target call ratio $r$
1: Initialize generated sequence $Y \leftarrow X$
2: Initialize the small-model KV cache $\mathcal{K}$ and logits $Z$ from $X$
3: Initialize the running call-threshold tracker $T$
4: **while** generation is not complete **do**
    *Update the threshold from the relative call-token score.*
5:     $a \leftarrow Z_{\text{CALL}} - \max_{j \neq \text{CALL}} Z_j$
6:     $T \leftarrow \text{UpdateTracker}(T, a, r)$
7:     $C \leftarrow (a \geq T.\text{threshold})$
    *Decode normally while excluding the call token.*
8:     $Z^{\neg\text{CALL}} \leftarrow \text{MaskCallLogit}(Z)$
9:     $v \leftarrow \text{Decode}(Z^{\neg\text{CALL}})$
    *Replace a call with continuation tokens from the cascade partner.*
10:     **if** $C$ **then**
      **v** *uses the small model's tokenizer, where the partner's output might map to multiple tokens*
11:        $\mathbf{v} \leftarrow \text{QueryPartner}(q, Y)$
12:        **for all** $v \in \mathbf{v}$ **do**
        *Append the final token and advance the small model state.*
13:          Append $v$ to $Y$
14:          $(Z, \mathcal{K}) \leftarrow p_\theta(v, \mathcal{K})$
15:        **end for**
16:     **else**
      *Append the final token and advance the small model state.*
17:        Append $v$ to $Y$
18:        $(Z, \mathcal{K}) \leftarrow p_\theta(v, \mathcal{K})$
19:     **end if**
20: **end while**
21: **return** $Y$

---

**Algorithm 3** Updating the running call threshold

---

**Require:** Tracker state $T$, current call advantage $a$, target call ratio $r$, smoothing coefficient $\alpha$
    *Update the recent history of call advantages.*
1: $T.\text{buffer} \leftarrow \text{UpdateRollingBuffer}(T.\text{buffer}, a)$
    *Estimate the threshold corresponding to the target call ratio.*
2: $\tau_{\text{target}} \leftarrow \text{Quantile}(T.\text{buffer}, 1 - r)$
    *Smooth the threshold across decoding steps.*
3: $T.\text{threshold} \leftarrow \alpha T.\text{threshold} + (1 - \alpha)\tau_{\text{target}}$
4: **return** $T$

---

## C Evaluation Tasks

**FactScore.** We evaluate factual precision using FactScore (Min et al., 2023), a benchmark for open-ended biography generation. Given a generated biography, FactScore uses GPT models to extracts a set of atomic facts and computes the proportion that is supported by a trusted knowledge source. We generate biographies for the 183 entities provided in the benchmark. We follow Zhao et al. (2025) in constructing a prompt template suitable for non-instruction-tuned models, given by `"Tell me a bio of <name>. <name> is"`. Factuality is validated using retrieval-augmented prompting with GPT 3.5 turbo (Min et al., 2023).

**NLU Tasks.** We use multiple-choice Natural Language Understanding tasks to evaluate out models both with and without cascading. We focus on benchmarks appropriate for small-scale models (Du et al., 2024): *ARC-Easy* (Clark et al., 2018), *HellaSwag* (Zellers et al., 2019), *PIQA* (Bisk et al., 2020) and *SIQA* (Sap

et al., 2019). Although *ARC-Easy* requires subject-level knowledge, we follow previous work (Zhao et al., 2025) in treating it as a general language benchmark rather than factual QA. We omit *Commonsense QA* (Talmor et al., 2019), as our models did not exceed chance-level performance on this benchmark. We use the eval-harness library. Models are evaluated in the standard way, comparing the loglikelihoods of the possible answers (A, B, C, etc).

**Factual Benchmarks.** We evaluate on two factual question answering tasks: *BigBench QA Wikidata* (Srivastava et al., 2022) and the long-tail subset of *PopQA* (Asai et al., 2024), which contains 1399 queries about rare entities (fewer than 100 monthly Wikipedia page views), Asai et al. (2023). Both tasks are evaluated with three shots. Performance is measured by checking whether the gold answer is contained in the model output. To help models understand the Q&A format, we provide 3 examples in front of each query. Furthermore, following Zhao et al. (2025), we prompt Gemini 2.0 Flash to rephrase the *PopQA* queries into a knowledge-completion task, which reduces the need for instruction-following ability. This way, the query `"What is Ufa the capital of?"` becomes `"What is Ufa the capital of?  Ufa is the capital of"`.

## D    Assessing Annotation Quality

In order to verify the quality of our spaCy-based factual annotations, we run a small-scale comparison of spaCy-based, LLM-judge-based (Zhao et al., 2025), and human annotations on two randomly selected training set examples. Human annotations were produced by the authors as a simple proxy for ground truth factuality. Due to the limitation in both scale and human annotation, these results should be taken as directional.

We measure precision and recall values treating the human annotation as ground truth, and publish the full annotations below in Section D.1. Table 6 shows that LaCy consistently achieves large F1 scores relative to the human annotation, outperforming LLM judge.

| Example | Metric | LMLM | spaCy |
|---|---|---|---|
| Burke Marshall | Precision | 77.8% | 87.1% |
| | Recall | 32.7% | 67.3% |
| | F1 | 46.1% | 75.9% |
| Jaguar XF | Precision | 96.4% | 83.1% |
| | Recall | 50.7% | 67.8% |
| | F1 | 66.5% | 74.7% |

Table 6: Comparison of LLM judge and spaCy-based factual annotations, relative to a ground truth human annotation.

**On measuring annotation quality.** However, we believe that annotation quality should be measured based on how much the model can utilize it to distinguish what it can learn. Part of our added rules were designed to incorporate the characteristics of AR models. For example, we consider only the first occurrences of factual tokens as facts, because a factual token only brings new information if it is not already in the context. Hence we believe that the most appropriate metric to judge the quality of our annotation is how much the model can take advantage of it, i.e. whether what we flagged as factual contributed to a model that calls better and has higher factual accuracy in the cascade. This is measured by the final FactScore of the LaCy or spaCy only-trained models. From our ablations Figure 6, spaCy only is on par with LLM-judge only annotation, while spaCy annotations are faster to compute.

### D.1    Sample Annotations

Green denotes facts. Please note that extra spaces in formatting are due to showing token boundaries.

**Burke Marshall - Human annotation.**

Burke Marshall ( October 1 , 1922 - June 2 , 2003 ) was an American lawyer and who served as the United States Assistant Attorney General for the Civil Rights Division during the Civil Rights Movement . Marshall was born in Plain field , New Jersey . He attended Phillips Exeter Academy , graduating in 1940 , and received a BA from Yale University in 1943 . He joined the army , working in the intelligence corps as a Japanese translator and crypto ana ly st . It was during his military service that he met Violet Person , whom he later married . After World War II , Marshall returned to Yale Law School , earning his LL. D. in 1951 ; he was admitted to the Washington , D. C. , bar the same year , joining the Washington - based law firm of Co v ington & Bur ling in 1952 , where he worked for ten years , specializing in anti trust law for clients such as Standard Oil . Marshall was appointed Assistant Attorney General in 1961 by Robert F. Kennedy , who was Attorney General in President John F. Kennedy ' s administration . Despite Marshall ' s lack of civil rights experience , he was put in charge of the Civil Rights Division . D uring his time in government , Marshall was a significant contributor to a number of advances in civil rights . In 1961 , segregation on inter state travel was banned . The following year , the University of Mississippi was forced to admit James Meredith , a well - qualified black student . Marshall and the Attorney General persuaded President Kennedy to enforce the order using federal troops . Marshall also ran a campaign to increase voter registration by black s . Within two years of coming into office , he had launched 42 federal lawsuit s against states to reform their electoral legislation . Marshall ' s focus was on results . He argued to not use the Four teen th Amendment to overcome discrimination , instead favor ing the federal government ' s constitutional power to regulate inter state commerce . As that power was reserved to the government , states had few legal options of re course . Marshall used it as a basis to write the 1964 Civil Rights Act , which prohibited discrimination in public facilities , in government , and in employment . Marshall resigned his office in December 1964 . President Lyn don B. Johnson wrote on Marshall ' s formal letter of resignation , "I have never known any person who rendered a better quality of public service." After leaving government , Marshall returned to commercial legal practice , briefly re join ing Co v ington and Bur ling before becoming a vice president and general counsel at IBM in 1965 .

**Burke Marshall - spaCy annotation.**

Burke Marshall ( October 1 , 1922 - June 2 , 2003 ) was an American lawyer and who served as the United States Assistant Attorney General for the Civil Rights Division during the Civil Rights Movement . Marshall was born in Plain field , New Jersey . He attended Phillips Exeter Academy , graduating in 1940 , and received a BA from Yale University in 1943 . He joined the army , working in the intelligence corps as a Japanese translator and crypto ana ly st . It was during his military service that he met Violet Person , whom he later married . After World War II , Marshall returned to Yale Law School , earning his LL. D. in 1951 ; he was admitted to the Washington , D. C. , bar the same year , joining the Washington - based law firm of Co v ington & Bur ling in 1952 , where he worked for ten years , specializing in anti trust law for clients such as Standard Oil . Marshall was appointed Assistant Attorney General in 1961 by Robert F. Kennedy , who was Attorney General in President John F. Kennedy ' s administration . Despite Marshall ' s lack of civil rights experience , he was put in charge of the Civil Rights Division . D uring his time in government , Marshall was a significant contributor to a number of advances in civil rights . In 1961 , segregation on inter state travel was banned . The following year , the University of Mississippi was forced to admit James Meredith , a well - qualified black student . Marshall and the Attorney General persuaded President Kennedy to enforce the order using federal troops . Marshall also ran a campaign to increase voter registration by black s . Within two years of coming into office , he had launched 42 federal lawsuit s against states to reform their electoral legislation . Marshall ' s focus was on results . He argued to not use the Four teen th Amendment to overcome discrimination , instead favor ing the federal government ' s constitutional power to regulate inter state commerce . As that power was reserved to the

government , states had few legal options of re course.  Marshall used it as a basis to write the 1964 Civil Rights Act , which prohibited discrimination in public facilities , in government , and in employment.  Marshall resigned his office in December 1964.  President Lyn don B. Johnson wrote on Marshall ' s formal letter of resignation , " I have never known any person who rendered a better quality of public service." After leaving government , Marshall returned to commercial legal practice , briefly re join ing Co v ington and Bur ling before becoming a vice president and general counsel at IBM in 1965.

### Burke Marshall - LMLM annotation.

Burke Marshall ( October 1 , 1922 – June 2 , 2003 ) was an American lawyer and who served as the United States Assistant Attorney General for the Civil Rights Division during the Civil Rights Movement.  Marshall was born in Plain field , New Jersey.  He attended Phillips Exeter Academy , graduating in 1940 , and received a BA from Yale University in 1943.  He joined the army , working in the intelligence corps as a Japanese translator and crypto ana ly st.  It was during his military service that he met Violet Person , whom he later married. After World War II , Marshall returned to Yale Law School , earning his LL. D. in 1951 ; he was admitted to the Washington , D. C. , bar the same year , joining the Washington – based law firm of Co v ington & Bur ling in 1952 , where he worked for ten years , specializing in anti trust law for clients such as Standard Oil.  Marshall was appointed Assistant Attorney General in 1961 by Robert F. Kennedy , who was Attorney General in President John F. Kennedy ' s administration.  Despite Marshall ' s lack of civil rights experience , he was put in charge of the Civil Rights Division.  D uring his time in government , Marshall was a significant contributor to a number of advances in civil rights.  In 1961 , segregation on inter state travel was banned.  The following year , the University of Mississippi was forced to admit James Meredith , a well – qualified black student.  Marshall and the Attorney General persuaded President Kennedy to enforce the order using federal troops. Marshall also ran a campaign to increase voter registration by black s.  Within two years of coming into office , he had launched 42 federal lawsuit s against states to reform their electoral legislation.  Marshall ' s focus was on results.  He argued to not use the Four teen th Amendment to overcome discrimination , instead favor ing the federal government ' s constitutional power to regulate inter state commerce.  As that power was reserved to the government , states had few legal options of re course.  Marshall used it as a basis to write the 1964 Civil Rights Act , which prohibited discrimination in public facilities , in government , and in employment.  Marshall resigned his office in December 1964.  President Lyn don B. Johnson wrote on Marshall ' s formal letter of resignation , " I have never known any person who rendered a better quality of public service." After leaving government , Marshall returned to commercial legal practice , briefly re join ing Co v ington and Bur ling before becoming a vice president and general counsel at IBM in 1965.

### Jaguar XF - Human annotation.

The Jaguar X F ( X 260 ) is an executive / mid – size luxury sports sal oon manufactured and marketed by the Jaguar Car s brand of Jaguar Land R over in sed an / sal oon and station wagon / e state body styles.  Following the first generation steel – bodied X 250 X F introduced in 2007 , the second – generation X F sed an / sal oon debuted at the 2015 New York International Auto Show , noted for its aluminium body work.  As of 2022 , the X F has been downgrade d to compete in the D – segment while retaining its E – segment exterior dimensions.  The X 260 X F uses 83 percent all – new parts compared with the previous model. The car is shorter than the predecessor.  Body work uses aluminium as the primary component of the body structure and chassis ; the X F ' s body side panel is a single aluminium pressing.  The chassis featured a fully independent suspension , including multiple " mode s " in the S model providing either maximum comfort , maximum performance , or a setting in

between.  The standard model in the USA featured a P 250247 HP In gen ium engine.  Optional power included the P 300296 HP turbo 4 and the P 380380 HP super charge d V 6 for the performance – oriented S model.  Diesel engines , while available in other markets , were not available to customers in North America.  The Sport b rake was introduced for sale in the USA in 2018 where it was positioned as a competitor for other high performance " station wagon s." It was initially available only in S trim for North America , with the high performance F – Type 3.  0 L super charge d 380 HP engine and all – wheel drive.  The X F S in sport b rake trim was able to accelerate from 0 – in 4.  9 seconds.  A long – wheel base version debuted at the 2016 Beijing Motor Show , with added to the wheel base giving rear passengers more leg room and more knee room.  As the car is designed for cha uff eur – driven drivers in mind , standard equipment includes folding tables , mass aging seats , electric window blind s , and eight – inch screens integrated in the back of the front – seat head rest s.  It is the first aluminium – bodied car built in China with the debut of some new features , with Clear Exit Detection warning passengers of opening the doors into traffic approaching from behind.  The X FL comes with a new cabin air ion isation technology to make the air inside the car more comfortable.  For the driver , the In Control Touch Pro info tainment system with its 10.  2 – inch touch screen works alongside a configurable 12. 3 – inch digital instrument cluster which works together with a 17 – s peak er , 825 – watt Mer id ian Sur round Sound System.  Rear passengers have access to a wide array of buttons as well as to 2 HDMI ports , 2 USB 3.  0 ports , and a 12 V power socket.  The 2.  0 – litre I 4 engine is offered in power output configurations and a 3.  0 – litre super charge d V 6 engine is also offered as the ultimate engine option , with all versions coming with an automatic transmission.  The long wheel base models come with four – wheel – drive only. Due to the long wheel base and added features , the X FL weigh s more than the standard X F.

### Jaguar XF - spaCy annotation.

The Jaguar X F ( X 260 ) is an executive / mid – size luxury sports sal oon manufactured and marketed by the Jaguar Car s brand of Jaguar Land R over in sed an / sal oon and station wagon / e state body styles.  Following the first generation steel – bodied X 250 X F introduced in 2007 , the second – generation X F sed an / sal oon debuted at the 2015 New York International Auto Show , noted for its aluminium body work.  As of 2022 , the X F has been downgrade d to compete in the D – segment while retaining its E – segment exterior dimensions.  The X 260 X F uses 83 percent all – new parts compared with the previous model. The car is shorter than the predecessor.  Body work uses aluminium as the primary component of the body structure and chassis ; the X F ' s body side panel is a single aluminium pressing.  The chassis featured a fully independent suspension , including multiple " mode s " in the S model providing either maximum comfort , maximum performance , or a setting in between.  The standard model in the USA featured a P 250247 HP In gen ium engine.  Optional power included the P 300 296 HP turbo 4 and the P 380380 HP super charge d V 6 for the performance – oriented S model.  Diesel engines , while available in other markets , were not available to customers in North America.  The Sport b rake was introduced for sale in the USA in 2018 where it was positioned as a competitor for other high performance " station wagon s." It was initially available only in S trim for North America , with the high performance F – Type 3.  0 L super charge d 380 HP engine and all – wheel drive.  The X F S in sport b rake trim was able to accelerate from 0 – in 4.  9 seconds.  A long – wheel base version debuted at the 2016 Beijing Motor Show , with added to the wheel base giving rear passengers more leg room and more knee room.  As the car is designed for cha uff eur – driven drivers in mind , standard equipment includes folding tables , mass aging seats , electric window blind s , and eight – inch screens integrated in the back of the front – seat head rest s.  It is the first aluminium – bodied car built in China with the debut of some new features , with Clear Exit Detection warning passengers of opening the doors into traffic approaching from behind.  The X FL comes with a new cabin air ion isation technology to make the air inside the car more comfortable. For the driver , the In Control Touch Pro info tainment system with its 10.  2 – inch

touch screen works alongside a configurable 12.  3 - inch digital instrument cluster which works together with a 17 - s peak er , 825 - watt Mer id ian Sur round Sound System.  Rear passengers have access to a wide array of buttons as well as to 2 HDMI ports , 2 USB 3.  0 ports , and a 12 V power socket.  The 2.  0 - litre I 4 engine is offered in power output configurations and a 3.  0 - litre super charge d V 6 engine is also offered as the ultimate engine option , with all versions coming with an automatic transmission.  The long wheel base models come with four - wheel - drive only.  Due to the long wheel base and added features , the X FL weigh s more than the standard X F.

## Jaguar XF - LMLM annotation.

The Jaguar X F ( X 260 ) is an executive / mid - size luxury sports sal oon manufactured and marketed by the Jaguar Car s brand of Jaguar Land R over in sed an / sal oon and station wagon / e state body styles.  Following the first generation steel - bodied X 250 X F introduced in 2007 , the second - generation X F sed an / sal oon debuted at the 2015 New York International Auto Show , noted for its aluminium body work.  As of 2022 , the X F has been downgrade d to compete in the D - segment while retaining its E - segment exterior dimensions.  The X 260 X F uses 83 percent all - new parts compared with the previous model. The car is shorter than the predecessor.  Body work uses aluminium as the primary component of the body structure and chassis ; the X F ' s body side panel is a single aluminium pressing.  The chassis featured a fully independent suspension , including multiple " mode s " in the S model providing either maximum comfort , maximum performance , or a setting in between.  The standard model in the USA featured a P 250247 HP In gen ium engine.  Optional power included the P 300296 HP turbo 4 and the P 380380 HP super charge d V 6 for the performance - oriented S model.  Diesel engines , while available in other markets , were not available to customers in North America.  The Sport b rake was introduced for sale in the USA in 2018 where it was positioned as a competitor for other high performance " station wagon s." It was initially available only in S trim for North America , with the high performance F - Type 3.  0 L super charge d 380 HP engine and all - wheel drive.  The X F S in sport b rake trim was able to accelerate from 0 - in 4.  9 seconds.  A long - wheel base version debuted at the 2016 Beijing Motor Show , with added to the wheel base giving rear passengers more leg room and more knee room.  As the car is designed for cha uff eur - driven drivers in mind , standard equipment includes folding tables , mass aging seats , electric window blind s , and eight - inch screens integrated in the back of the front - seat head rest s.  It is the first aluminium - bodied car built in China with the debut of some new features , with Clear Exit Detection warning passengers of opening the doors into traffic approaching from behind.  The X FL comes with a new cabin air ion isation technology to make the air inside the car more comfortable.  For the driver , the In Control Touch Pro info tainment system with its 10.  2 - inch touch screen works alongside a configurable 12. 3 - inch digital instrument cluster which works together with a 17 - s peak er , 825 - watt Mer id ian Sur round Sound System.  Rear passengers have access to a wide array of buttons as well as to 2 HDMI ports , 2 USB 3.  0 ports , and a 12 V power socket.  The 2.  0 - litre I 4 engine is offered in power output configurations and a 3.  0 - litre super charge d V 6 engine is also offered as the ultimate engine option , with all versions coming with an automatic transmission.  The long wheel base models come with four - wheel - drive only. Due to the long wheel base and added features , the X FL weigh s more than the standard X F.

# E    Discussion on Domain Dependence

Our notion of acceptability generalizes across domains: the universally relevant question for delegation is not whether a model reproduces the exact next token in the dataset, but whether its continuation remains semantically and factually valid. The main claim of this work is therefore not specific to Wikipedia or to spaCy-based factual annotations. Rather, we claim that token-level cross-entropy loss alone is not always a sufficiently precise proxy for this objective.

**Why Wikipedia is suitable to study delegation.**    How acceptability can be approximated, and how useful it is to do so may depend strongly on the domain. We intentionally study Wikipedia-style factual generation because it provides a controlled setting in which acceptability violations can be evaluated precisely and at fine granularity using established metrics such as FactScore (Min et al., 2023). This aligns with prior work on factual delegation and externalized knowledge, where Wikipedia-based generation has become a standard testbed (Zhao et al., 2025). A key property of this setting is that factual information is often localized. Entity names, dates, occupations, locations, and similar information often appear as isolated factual units whose correctness can be evaluated independently from the surrounding text. This makes Wikipedia-style factual knowledge particularly suitable for studying token-level delegation decisions: certain pieces of information can be cleanly externalized, while the remaining language modeling problem remains locally coherent. Our spaCy-based factual annotations should therefore be interpreted as an approximation to acceptability appropriate for domains with isolated factual units.

**Transfer to other domains**

**Appoximating acceptability.**    For other domains, the acceptability analysis introduced in Section 3 can be repeated to identify which token types or structures are most associated with harmful errors and hence are most useful for delegation decisions. However, we do not expect the wikipedia-specific approximation to transfer unchanged to all domains. In reasoning-heavy domains or tasks such as mathematics, theorem proving, or code generation, correctness often depends less on localized factual units and more on global compositional structure. Mistakes may be harder to attribute to specific tokens. In such settings, the relevant notion of acceptability may depend on coarser structures such as concepts (Shani et al., 2023) or even properties of the entire document. Therefore, different auxiliary signals may be required to approximate acceptability effectively.

**Differences in retrieval efficiency.**    When information is not localized to tokens, it becomes more difficult to retrieve efficiently. Retrievals of multiple tokens might be necessary, which can result in a mismatch between the SLM and cascade partner, for example, the cascade partner proposing a difficult mathematical argument the SLM is unable to continue correctly. Therefore, retrieval of full documents or problems may be a more feasible approach for reasoning-heavy domains.

# F  Additional Results

## F.1  Extended Acceptability Analysis

**Early Training.**  After 10B tokens only, the difference in the acceptability of factual and non-factual tokens is already visible (Figure 8).

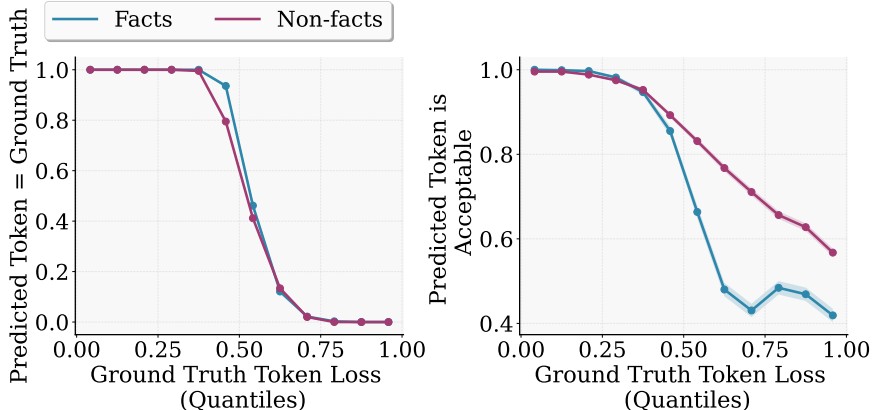

Figure 8: **The difference between Accuracy and Acceptability after training on 10B tokens.** The token loss is predictive of whether a token is likely to match its exact ground-truth token *(left)*. However, this signal is blind to the type of token: Non-factual tokens are considered equally wrong as factual tokens, although non-factual tokens with high loss often do not render an output false *(right)*. We utilize a SpaCy grammar parser during pretraining to tell these two signals apart. Results are based on 44k tokens, Standard Errors shown.

**Late Training.**  After 50B steps, and averaging over 100k tokens standard errors around the mean are barely visible (Figure 9).

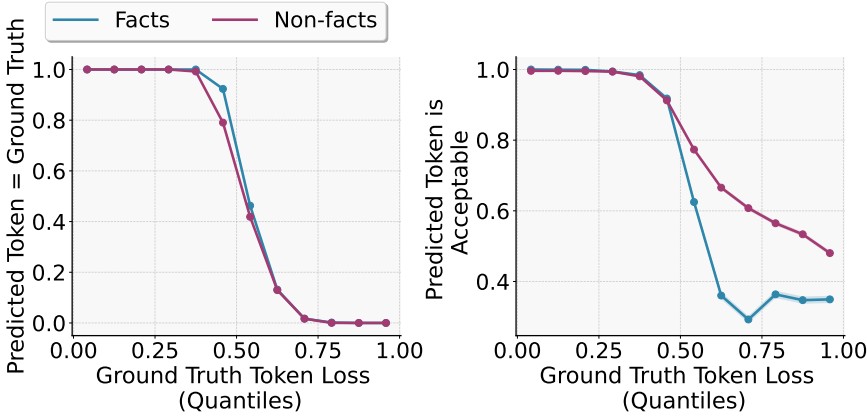

Figure 9: **The difference between Accuracy and Acceptability after training on 50B tokens (as shown in Figure 2 in the main text).** The token loss is predictive of whether a token is likely to match its exact ground-truth token *(left)*. However, this signal is blind to the type of token: Non-factual tokens are considered equally wrong as factual tokens, although non-factual tokens with high loss often do not render an output false *(right)*. We utilize a SpaCy grammar parser during pretraining to tell these two signals apart. Results are based on 100k tokens, Standard Errors are shown

## F.2 Main Results: FactScore and Fact Leakage

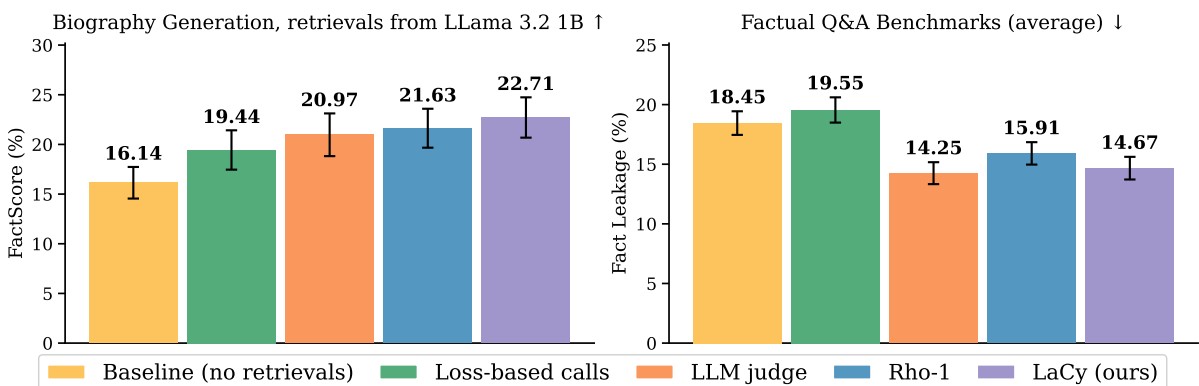

Figure 10: **Results overview for pretraining a 334M SLM.** *(Left.)* The LaCy-trained SLM achieves the highest FactScore when generating biography with Llama 3.2 1B as cascade partner, confirming that it successfully generates calls at factual token positions. *(Right.) Without calling*, LaCy has lowest fact leakage, meaning the least facts were trained into the limited parametric SLM memory. Error bars show 95% bootstrap confidence intervals (10,000 resamples).

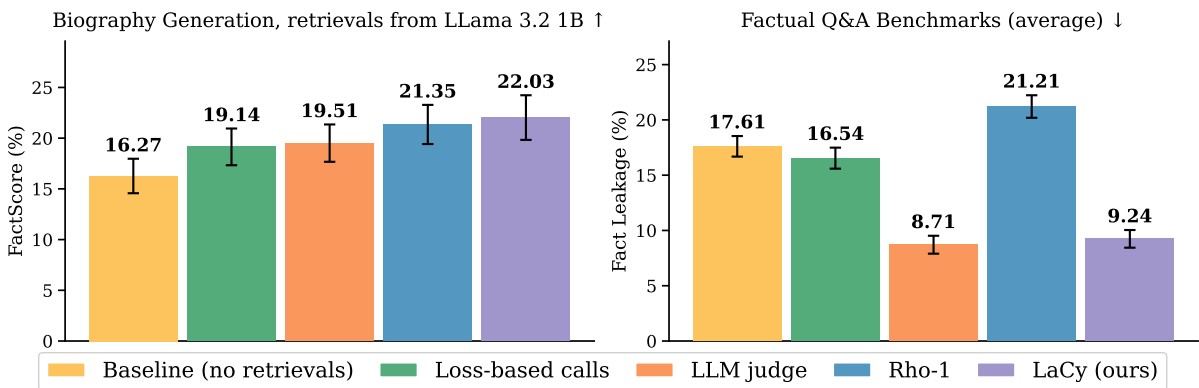

Figure 11: **Results overview for pretraining a 1.3B SLM.** *(Left.)* The LaCy-trained SLM achieves the highest FactScore when generating biography with Llama 3.2 1B as cascade partner, confirming that it successfully generates calls at factual token positions. *(Right.) Without calling*, LaCy has low fact leakage, meaning the least facts were trained into the limited parametric SLM memory. Error bars show 95% bootstrap confidence intervals (10,000 resamples).

### F.3 Results in the RAG Setup

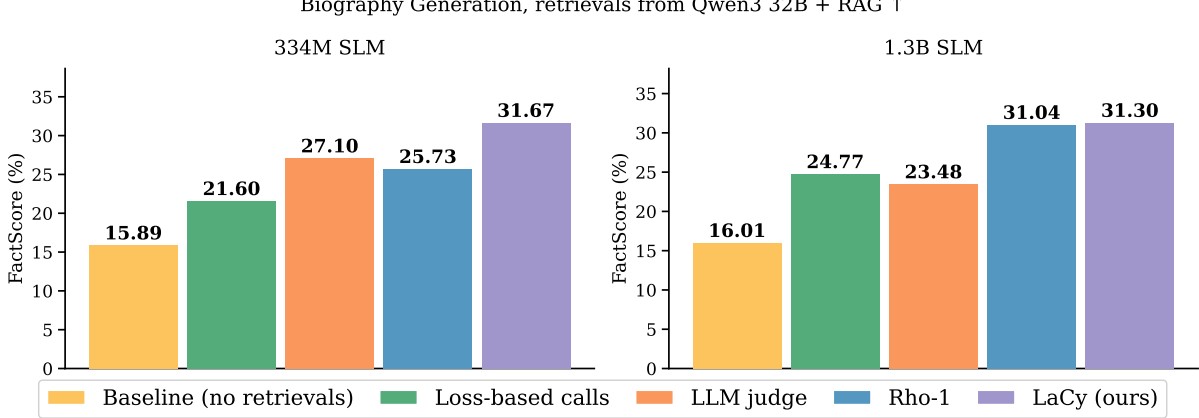

Figure 12: **FactScore results using RAG-enchanced Qwen 3 32B as cascade partner.** *(Left.)* 334M parameter SLM. *(Right.)* 1.3B SLM. The LaCy-trained SLM achieves the highest FactScore when generating biography with RAG-enhanced Qwen 3 32B as cascade partner, confirming that it successfully generates calls at factual token positions.

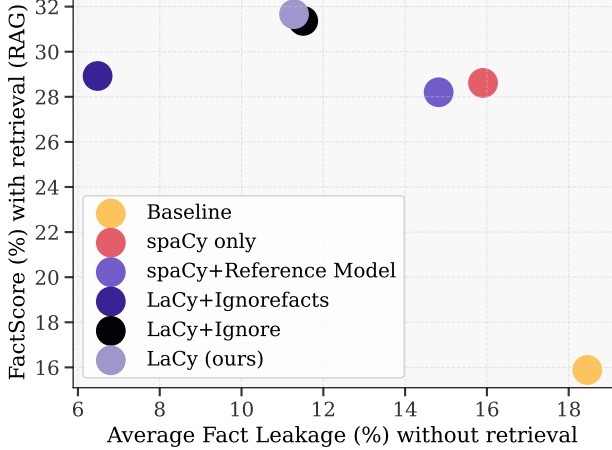

Figure 13: **FactScore (with cascade) against fact leakage (without cascade) for LaCy ablations in the RAG setup, for 334M parameter SLMs.** Methods disabling backpropagation on $x\%$ tokens are evaluated after $x\%$ more training steps. Loss signal is beneficial: spaCy (without loss) performs worse than LaCy. Using a reference loss or ignoring non-delegated facts does not give improvements on FactScore despite computational overhead (Table 2). Offloading even more tokens (LaCy+Ignore) is not beneficial.

### F.4 Comparison to Off-the-shelf Baselines

To understand how LaCy's improvement of factuality fares relative to models with larger baseline capacity, we compare LaCy to off-the-shelf baselines and LMLM (Zhao et al., 2025).

Table 7 shows the results. Off-the-shelf models are marked in gray, and "*" denotes results obtained in Zhao et al. (2025). We see that 334M LaCy outperforms Llama 2-7B, a more than 20 times larger model.

Comparison to LMLM (Zhao et al., 2025) is more subtle due to the following two crucial differences (the FactScore evaluation protocol is identical):

- LMLM models are trained for fewer steps (corresponding to about 25B tokens compared to our 50B), with a larger vocabulary size
- LMLM employs database retrievals in which the number of retrievals are not constrained, and we expect them to be higher than the train-time delegation ratio due to an added bias to logits.

Hence the "LMLM + database" entry is not directly comparable and is only reported for context. The rigorous apples-to-apples comparison with LMLM is LLM judge. The benefit of LMLM over LLM judge comes from the retrieval setup (database retrievals are correct by definition).

| Method | Model Size | FactScore (%) |
|---|---|---|
| baseline | 334M | 15.89 |
| baseline | 1.3B | 16.01 |
| LaCy | 334M | 22.71 |
| LaCy (RAG) | 334M | 31.67 |
| LaCy | 1.3B | 22.03 |
| LaCy (RAG) | 1.3B | 31.30 |
| LMLM baseline* | 355M | 14.40 |
| LMLM + database* | 355M | 23.90 |
| Llama 3.2 1B (Meta AI, 2024) | 1B | 34.20 |
| Qwen 3 32B + RAG (Qwen Team, 2025) | 32B | 79.00 |
| OPENAI/GPT2* (Radford et al., 2019) | 774M | 17.40 |
| Llama 2-7B* (Team, 2023) | 7B | 21.10 |
| Llama 3.1-8B* (Team, 2024) | 8B | 40.30 |

Table 7: FactScore comparison across methods and model sizes. Off-the-shelf baselines are marked in gray, "*" denotes results taken from prior work with identical evaluation protocol (Zhao et al., 2025)

### F.5 Ablations

### F.5.1 Delegation Method Ablations after Equalized 400k Training Steps

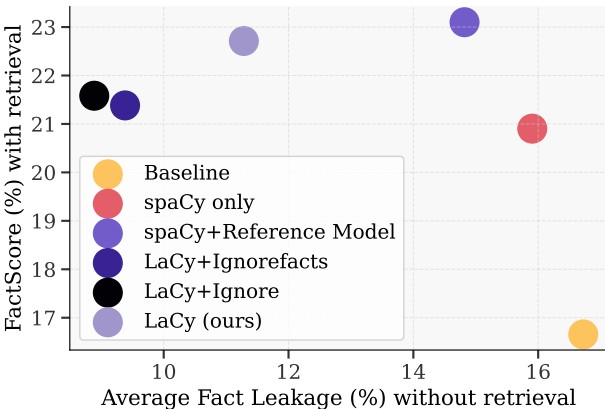

Figure 14: **FactScore (with cascade) against fact leakage (without cascade) for LaCy ablations, evaluated at an equalized number of 400k training steps.** Loss signal is beneficial: spaCy (without loss) performs worse. Using a reference loss or ignoring non-delegated facts gives marginal improvements on FactScore at a computational overhead (Table 2). Offloading even more tokens is no longer beneficial once training steps are equalized.

### F.5.2 Analysis of Token Selection Overlap

In order to further disentangle the effects of spaCy factuality annotations and loss-based difficulty signals, we analyze the overlap of tokens selected for delegation by various methods. Our analysis scores $\approx 1.2$B tokens from the dwiki validation set. For each selection method, we record the selection mask $M$ and compute the aggregate value of $|M \cap \text{LaCy}|/|\text{LaCy}|$, where "LaCy" stands for the LaCy selection mask. This corresponds to the recall with respect to LaCy. Note that due to the equal call budget for each method, all masks select the same amount of tokens to delegate, hence $|M| = |\text{LaCy}|$. Therefore, the precision $|M \cap \text{LaCy}|/|M|$ is automatically equal to the recall.

In Table 8 we present precision and recall values for 15% and 10% delegation budgets. In the 10% delegation budget, we include further, hybrid methods such as LLM judge + loss and LLM judge + rholoss. These cannot be tested at 15% budget, because there are only 15% LLM-judge-annotated facts. Hence we would have to select all of them at a 15% and cannot use the loss or rholoss signal for further filtering. The methods spaCy only and LLM jduge at 10% budget sample delegations randomly of the fact pool. Results in Table 8 for these methods are averaged over 8 random seeds.

**LaCy components.** Selecting highest-loss tokens achieves larger precision / recall, especially at a 10% call budget. Our ablation on spaCy only in Figure 6 and Figure 14 shows that spaCy only is on-par with LLM judge on FactScore. Together, these findings show that the loss signal is essential, but not the only driver of LaCy selection, and in terms of the quality of factuality signal, LLM judge and spaCy are comparable, with spaCy being cheaper.

**Hybrid factuality + loss methods.** The hybrid method spaCy + rholoss has 80-90% overlap with LaCy selection, and therefore we suspect would perform similarly to LaCy. LLM judge-based hybrids produce generally distinct selection masks at the 10% delegation budget. This is largely because spaCy has about

27% calls where LLM judge has only 15%, hence the flexibility of loss-based selection is smaller. We therefore expect these methods to perform closer to LLM judge on FactScore.

| Method | Budget | Overlap with LaCy |
|---|---|---|
| spaCy only | 15% | 54.6% |
| 15% highest loss | 15% | 62.9% |
| spaCy + rholoss | 15% | 92.4% |
| LLM judge | 15% | 40.9% |
| spaCy only | 10% | 36.4% |
| 10% highest loss | 10% | 68.7% |
| spaCy + rholoss | 10% | 81.3% |
| LLM judge | 10% | 29.5% |
| LLM judge + loss | 10% | 41.0% |
| LLM judge + rholoss | 10% | 31.0% |

Table 8: Overlap of each method's selected tokens with LaCy (precision/recall, which are equal due to fixed delegation budget). The horizontal rule separates the 15% and 10% delegation budget analyses. The methods spaCy only and LLM jduge at 10% budget sample delegations randomly of the fact pool. Results for these methods are averaged over 8 random seeds.

**Qualitative Analysis** We supply examples of the first 25 delegated tokens in the biography of Burke Marshall (see the corresponding Wikipedia article among the sample annotations in Section D.1). Tokens are separated by "·", whitespace tokens are denoted by [sp].

At a 15% budget, the LaCy and spaCy + rholoss selected sets are equal. Compared to spaCy only and LLM judge, LaCy or factuality + loss methods tend to delegate only parts of dates, whilehigh loss only delegates many space tokens.

```
LaCy (15%): first 25 selected tokens for Burke Marshall.

Burke · Marshall · October · 2 · June · 2 · 0 · 3 · United · Assistant · Civil · Civil · Movement
· Plain · Phillips · Exeter · BA · Yale · 3 · Japanese · Violet · Person · World · Yale · Law
```

```
15% highest loss: first 25 selected tokens for Burke Marshall.

Burke · Marshall · October · 2 · June · 2 · 0 · 3 · who · Assistant · Civil · Civil · [sp] · [sp] ·
[sp] · [sp] · Plain · Phillips · Exeter · BA · Japanese · It · Violet · Person · [sp]
```

```
spaCy only: first 25 selected tokens for Burke Marshall.

Burke · October · 1 · 9 · 2 · 2 · 2 · American · States · Assistant · Attorney · Civil · Movement
· New · Phillips · 9 · BA · Yale · 1 · 4 · 3 · Japanese · Violet · World · 1
```

```
spaCy + rholoss: first 25 selected tokens for Burke Marshall.

Burke · Marshall · October · 2 · June · 2 · 0 · 3 · United · Assistant · Civil · Rights · Civil ·
Movement · Plain · Phillips · Exeter · BA · Yale · 3 · Japanese · Violet · Person · World · Yale ·
Law
```

```
LLM judge: first 25 selected tokens for Burke Marshall.

June · 2 · 2 · 0 · 0 · 3 · American · lawyer · United · States · Assistant · Attorney · General
· for · Civil · Rights · Division · Plain · field · New · Jersey · Phillips · Exeter · Academy ·
Yale
```

> **LaCy (10%): first 25 selected tokens for Burke Marshall.**
>
> Burke · Marshall · October · 2 · June · 2 · 0 · 3 · Assistant · Civil · Civil · Plain · Phillips · Exeter · BA · Japanese · Violet · Person · World · 5 · 1 · Washington · Co · Bur · ten · Standard · Robert · Attorney · President · University · James · Meredith · President · two

> **LLM judge + loss (10%): first 25 selected tokens for Burke Marshall.**
>
> June · 2 · 2 · 0 · 3 · lawyer · United · Assistant · Attorney · for · Civil · Rights · Division · Plain · field · New · Jersey · Phillips · Exeter · Academy · Yale · University · Yale · Law · Washington

> **LLM judge + rholoss (10%): first 25 selected tokens for Burke Marshall.**
>
> June · 2 · 2 · 0 · 3 · American · lawyer · United · States · Assistant · Attorney · General · for · Civil · Division · Plain · field · New · Jersey · Phillips · Exeter · Academy · Yale · University · Yale

### F.5.3   Analysis of How LaCy Handles Multi-token Facts

In this section, we test whether token-level delegation causes LaCy models to miss important delegations when facts consist of multiple tokens. We conduct a statistical analysis on the 334m LaCy-trained model's biography generations used for FactScore. We used our spaCy parser to annotate the generated text for factuality. This gives us a measure of ground truth factuality for the model's generation: ideally, most facts in the generation should have come from the cascade partner.

We specifically examined multi-token facts, which are facts flagged by spaCy that map to multiple tokens when tokenized. We then counted, for every multi-token fact, whether LaCy called on one or multiple tokens corresponding to the fact. Since some facts may contain less truth-critical tokens, such as "in" and "the" in "actress in the West End theatre", we do not require the SLM to always call on every single token within the fact. We find that 56.93'% of multi-token facts encountered multiple calls, while 30.52% encountered only one, and 12.55% encountered zero calls.

The results show that LaCy encounters limitations in factual segmentation: it can miss to detect full facts or parts of facts. However, some of this is expected, because we did not train LaCy to delegate all facts, but only high loss facts. Hence we expect LaCy-trained SLMs to learn some facts themselves, removing the necessity to call on them. Our Figure 4 illustrates this point: for example, LaCy knows "Hood" after seeing "Robin". In addition, part of these inaccuracies can step from cases where the generated sequence could be followed by a fact "Errol Flynn [...] appeared in", but the cascade partner may return a non-factual token, e.g. "more".

### F.5.4   The Impact of *How Much* LaCy Calls

**Pre-training-time Calls.**   We retrain all methods in our main results on 10% train-time calling ratio. Figure 15 and Figure 16 show similar trends to the main results: LaCy consistently outperforms other methods on FactScore, and almost always on fact leakage.

**Inference-time Calls (Table 9)**   We ablate the calling ratio in trained LaCy models at inference time. We report the empirically observed calling ratios, when setting target ratios $0\%, 15\%, 45\%, 60\%$ and $75\%$. Results reveal that delegation is helpful, but only marginally. For example, a $+60\%$ delegation ratio results only in an around 2% increase for 334m and 6% for 1.3B.

### F.5.5   The Impact of in *Which Training Stage* LaCy Calls

Delegation during pre-training requires adapting existing training setups, which can be cumbersome. We therefore study whether LaCy's benefits on FactScore can be replicated through inference-time only delegation.

334m model, 10% call ratio

Figure 15: **Results for pretraining a 334M SLM with 10% calling ratio.** *(Left.)* The LaCy-trained SLM achieves the highest FactScore when generating biography with Llama 3.2 1B as cascade partner, confirming that it successfully generates calls at factual token positions. *(Right.) Without calling*, LaCy has low fact leakage, meaning that few facts were trained into the limited parametric SLM memory.

1.3B model, 10% call ratio

Figure 16: **Results for pretraining an 1.3B SLM with 10% calling ratio.** *(Left.)* The LaCy-trained SLM achieves the highest FactScore when generating biography with Llama 3.2 1B as cascade partner, confirming that it successfully generates calls at factual token positions. *(Right.) Without calling*, LaCy has lowest fact leakage, meaning the least facts were trained into the limited parametric SLM memory.

Note that the fact leakage evaluation tests SLMs without delegation, hence delegation time does not play a role in that experiment.

**Inference-time-only LaCy (i-LaCy).** We propose inference-time-only version of LaCy (i-LaCy), which adapts the spaCy and loss components as follows:

- **spacy:** we use the spaCy annotator *on the fly* during generation, to judge the factuality of the proposed (highest-probability) next token of the base model (SLM). This differs from the train-time method which decides factuality of the true next token.

- **loss**: as a replacement to the cross-entropy loss, we compute one of two uncertainty metrics:
    - the negative log probability of the highest-probability next token
    - the entropy of the normalized distribution of the top 20 next-tokens

| Model Size | Empirical Calling Ratio (%) | FactScore (%) |
|---|---|---|
| 334M | 0.00 | 15.22 |
| 334M | 22.40 | 22.71 |
| 334M | 40.35 | 23.07 |
| 334M | 60.20 | 23.77 |
| 334M | 79.73 | 25.56 |
| 1.3B | 0.00 | 17.41 |
| 1.3B | 22.52 | 22.03 |
| 1.3B | 40.41 | 23.62 |
| 1.3B | 60.40 | 25.63 |
| 1.3B | 80.29 | 28.65 |
| Llama 3.2 1B | 100.00 | 34.21 |

Table 9: **Impact of the Inference-time Call Ratio on LaCy.** Delegation is helpful, but only marginally: +60% delegation results only in an around 2% increase for 334m and 6% for 1.3B.

The above two signals are combined similarly to LaCy: only those positions can be delegated, where the SLM would predict a fact.

**Setup and Hyperparameters.**    We use our vanilla baseline as the base SLM. There are no modifications to standard pre-training, as delegations are only made at inference time. Retrievals use Llama 3.2 1B, consistent with the main results, and the same inference hyperparameters. In particular, we enforce a call rate of approximately 22%. <CALL> rates are managed by the same tracker. The SLM's factual predictions are replaced by a <CALL> based on thresholding the uncertainty metrics, and dynamic thresholds are computed in the same way as before.

**Baseline.**    We provide an inference-time-only random baseline that delegates randomly at each generation step, with probability 0.22.

**Results (Table 10).**    Random delegation surpasses the baseline without delegations on 334M (see Figure 10), but falls slightly below at 1.3B (Figure 11). Random inference-time delegation consistently stays below loss-based and fact annotation-based delegation methods.

i-LaCy improves over random delegation, but FactScorez are worse than LaCy in all configurations evaluated. We hypothesize that this is because the LaCy base model was trained on less facts, hence it generates less facts by itself. Thus, at the same call rate, less incorrect facts make it to the cascaded generation. The negative log probability and entropy seem to be equally good measures of internal uncertainty in the baseline model's factual next-token predictions.

**Conclusion.**    While i-LaCy does not reach LaCy's FactScore performance, it can be used as a simple adjustment to existing decoding methods for delegation, including for pre-trained models, or whenever the pre-training process cannot be modified.

| Model Size | Delegation Method | FactScore (%) |
|---|---|---|
| 334m | random | 19.00% |
| 334m | i-LaCy (log prob) | 20.51% |
| 334m | i-LaCy (entropy) | 20.17% |
| 1.3B | random | 15.50% |
| 1.3B | i-LaCy (log prob) | 20.16% |
| 1.3B | i-LaCy (entropy) | 20.26% |

Table 10: **FactScore Evaluation with Inference-time-only Delegation.** We replicate LaCy in an inference-time only setup, and confirm that pre-training-time delegation is *better* (LaCy achieves 22.71% FactScore at 334M and 22.03% at 1.3B scale).

### F.6   Full Results on NLU Performance

Table 11: **NLU performance of `<CALL>` augmented models, including ablations, *without* cascade.** We confirm that offloading facts only does not significantly degrade Natural Language Understanding (NLU). However, ignoring more tokens (as in Lacy+Ignore) harms NLU performance.

| Model | Metrics | | | | |
|---|---|---|---|---|---|
| | ARC Easy | HellaSwag | PIQA | SIQA | Average |
| Random chance | 25.0 | 25.0 | 50.0 | 33.3 | 33.3 |
| Baseline | 34.8 | **28.8** | 59.0 | 35.9 | 39.6 |
| Loss-based calling | 34.3 | 28.6 | 57.1 | 36.3 | 39.1 |
| Rho-1 | 35.0 | 28.6 | 56.8 | 35.9 | 39.1 |
| LLM judge | 33.8 | 28.3 | 57.3 | **36.8** | 39.1 |
| LaCy | **35.6** | 28.5 | **59.3** | 36.2 | **39.9** |
| LaCy + Reference loss | 34.8 | 28.6 | 57.1 | 35.7 | 39.1 |
| LaCy + Ignorefacts | 34.0 | 28.7 | 57.4 | 35.9 | 39.0 |
| LaCy + Ignore | 30.8 | 27.6 | 55.1 | 34.3 | 37.0 |

### F.7   Full Results on Loss vs FactScore

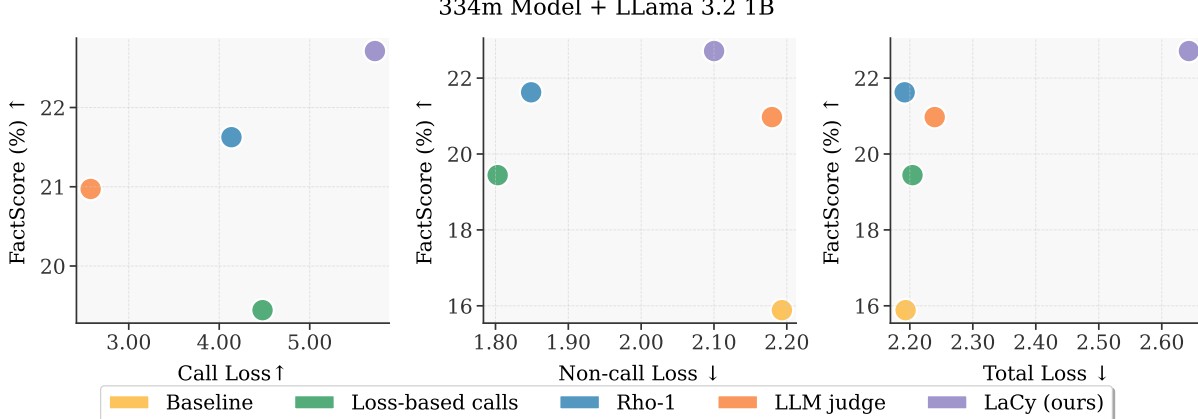

Figure 17: **Validation loss is not correlated with FactScore, 334M SLMs.** Neither the call loss *(Right)*, non-call loss *(Middle)*, nor the total loss *(Left)* is predictive of the FactScore of the displayed methods. Findings linking loss with downstream performance in related work Kaplan et al. (2020); Srivastava et al. (2022); Krajewski et al. (2025). do not transfer to our token-selection setting.

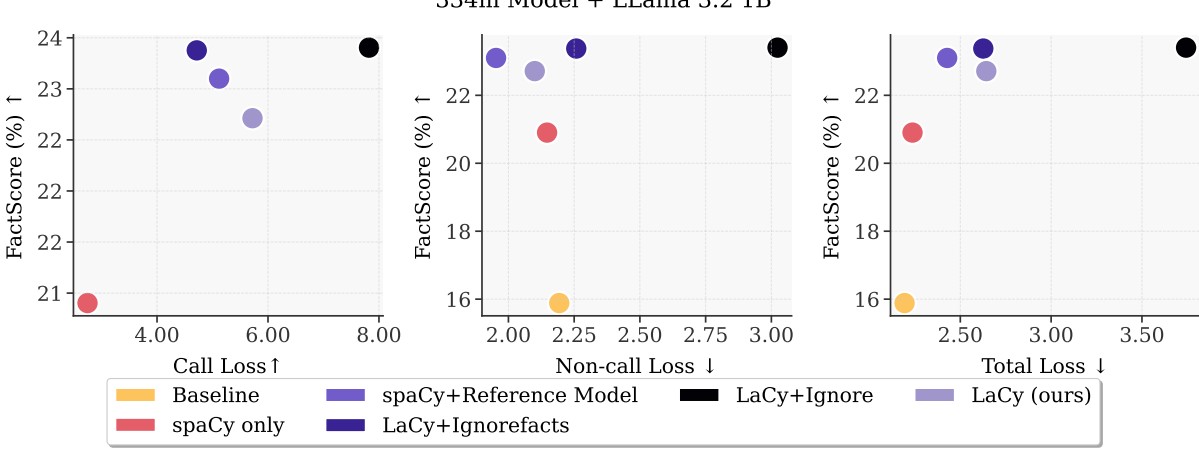

Figure 18: **Validation loss is not correlated with FactScore, 334M LaCy ablations.** Neither the call loss *(Right)*, non-call loss *(Middle)*, nor the total loss *(Left)* is predictive of the FactScore of the displayed methods. Findings linking loss with downstream performance in related work Kaplan et al. (2020); Srivastava et al. (2022); Krajewski et al. (2025). do not transfer to our token-selection setting.

## F.8 Comparison of Validation Losses

For each `<CALL>`-augmented method, we construct its call mask by selecting the top 15% call logits in a batch. Full colors show the loss values of the `<CALL>`-augmented methods, while light colors show the loss of a vanilla baseline evaluated on the *same* `<CALL>` mask. Across 334M and 1.3B parameter scales, LaCy calls on high-loss tokens (baseline call loss is high), and learns even less about them. Its non-call loss is competitive with the factuality-based LLM judge.

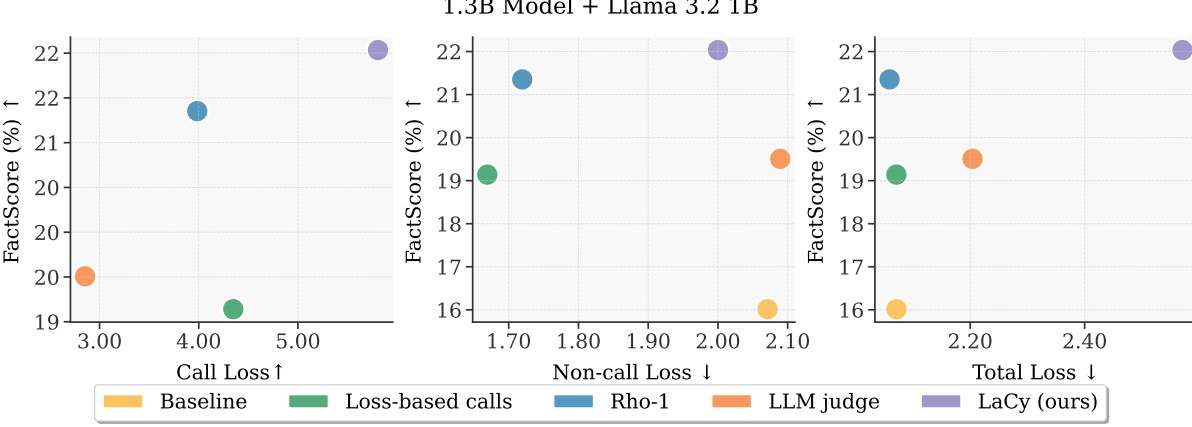

Figure 19: **Validation loss is not correlated with FactScore, 1.3B SLMs.** Neither the call loss *(Right)*, non-call loss *(Middle)*, nor the total loss *(Left)* is predictive of the FactScore of the displayed methods. Findings linking loss with downstream performance in related work Kaplan et al. (2020); Srivastava et al. (2022); Krajewski et al. (2025). do not transfer to our token-selection setting.

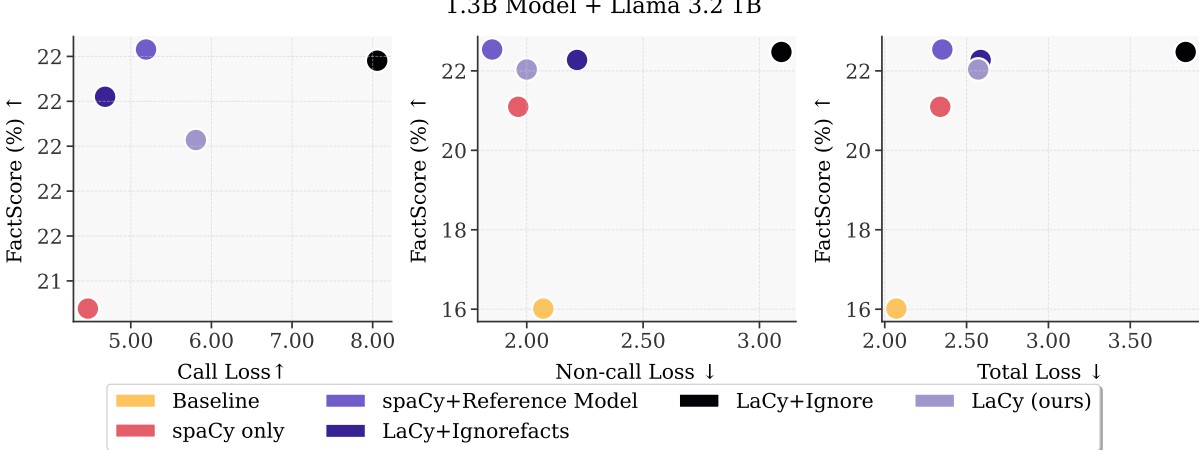

Figure 20: **Validation loss is not correlated with FactScore, 1.3B LaCy ablations.** Neither the call loss *(Right)*, non-call loss *(Middle)*, nor the total loss *(Left)* is predictive of the FactScore of the displayed methods. Findings linking loss with downstream performance in related work Kaplan et al. (2020); Srivastava et al. (2022); Krajewski et al. (2025). do not transfer to our token-selection setting.

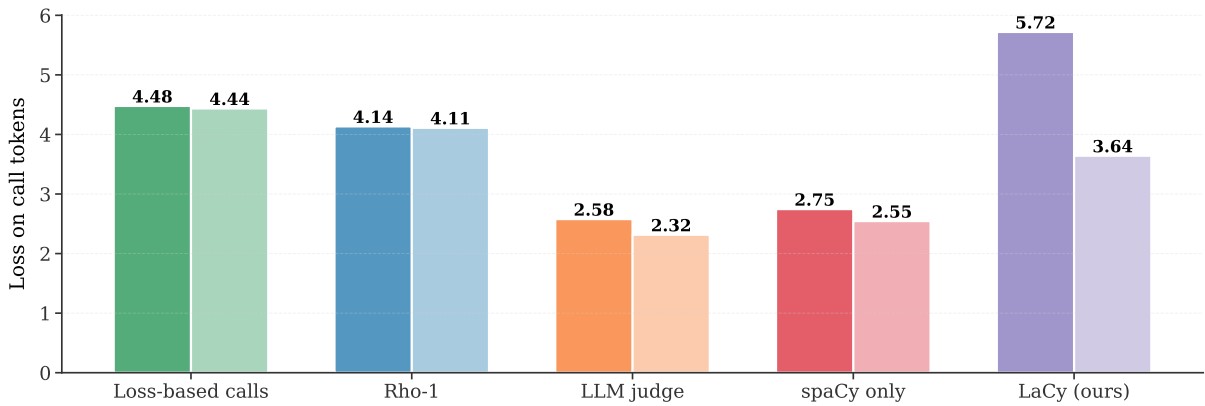

Figure 21: Call loss comparison for 334M parameter models

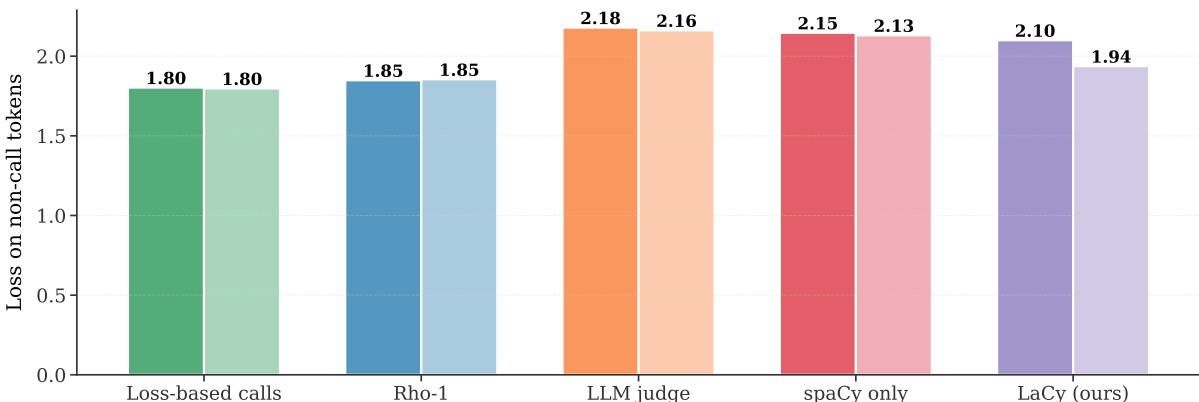

Figure 22: Non-call loss comparison for 334M parameter models

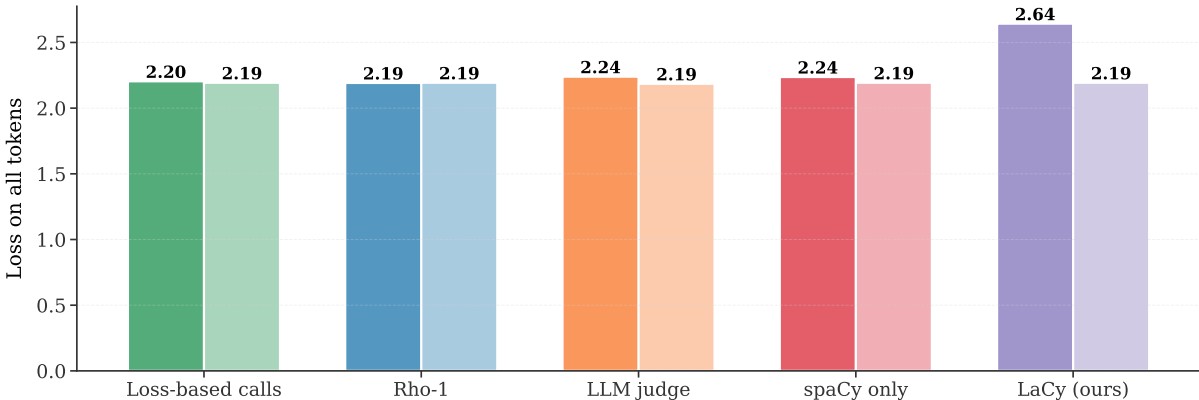

Figure 23: Total loss comparison for 334M parameter models

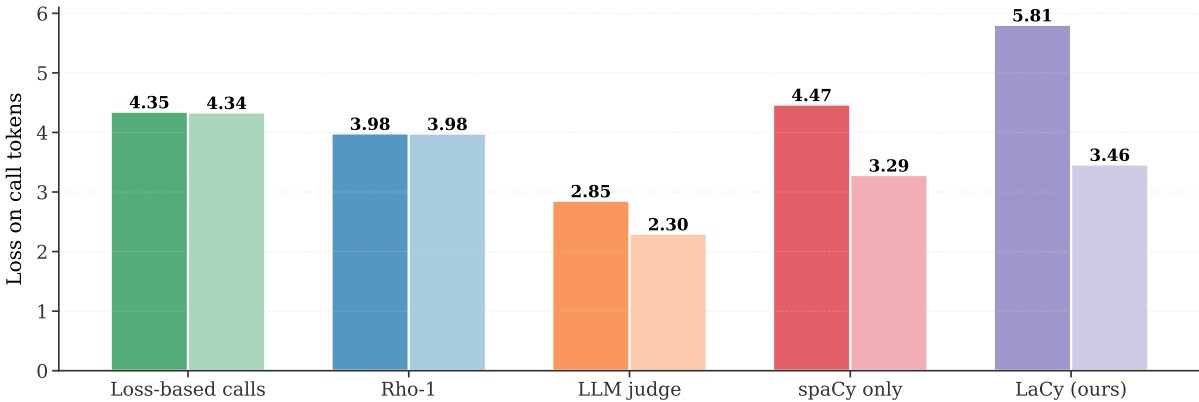

Figure 24: Call loss comparison for 1.3B parameter models

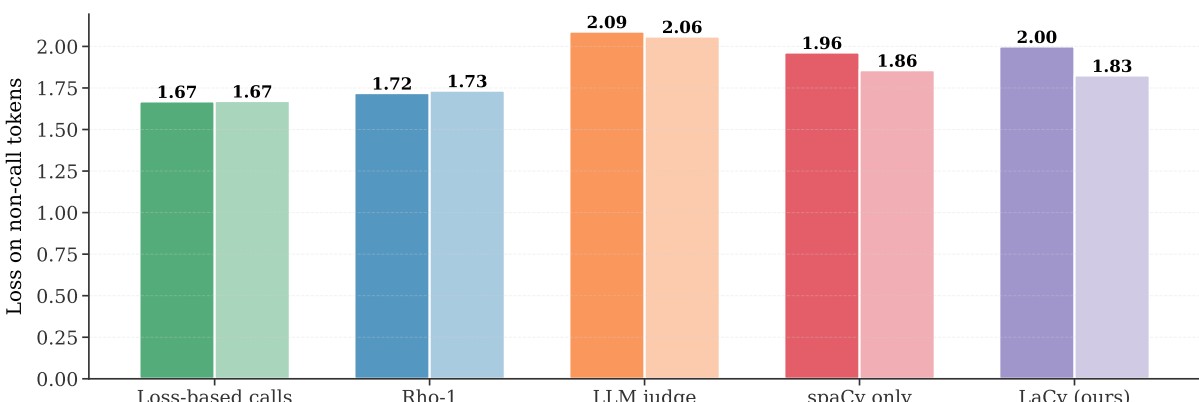

Figure 25: Non-call loss comparison for 1.3B parameter models

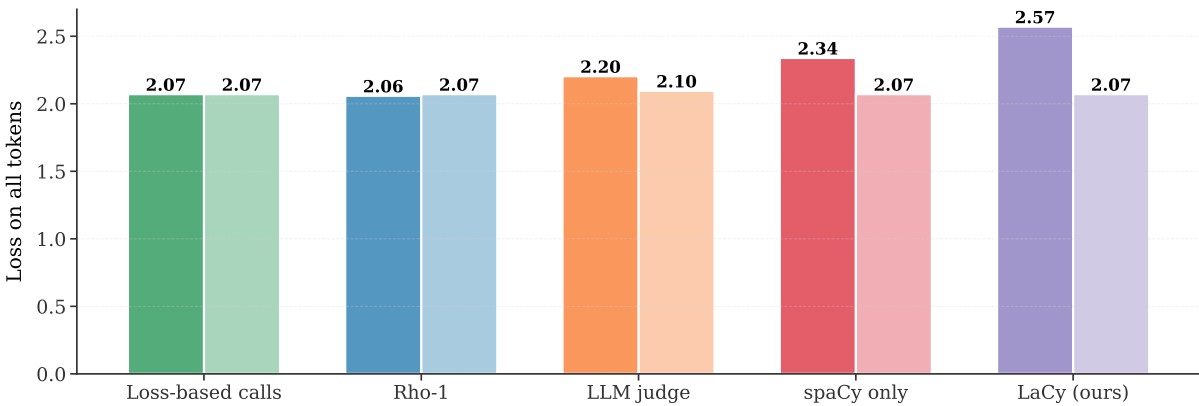

Figure 26: Total loss comparison for 1.3B parameter models

