# OpenReview forum: "LaCy: What Small Language Models Can and Should Learn is Not Just a Question of Loss"
_TMLR — Under review for TMLR_

### Review · Reviewer_bfXs · 2026-06-06

**Summary Of Contributions:**

The paper looks at an important question for Small Language Models which information should the model learn during pretraining, and which information should be obtained through external help such as retrieval systems or larger language models.
The authors argue that using standard cross entropy loss alone is not enough to decide when an SLM should ask for external help, because tokens with high loss are not always important for factual accuracy. To address this, they propose LaCy, a training method that combines loss signals with lightweight factual annotations generated using spaCy. These annotations help identify tokens that should trigger special call tokens for external assistance, and these call tokens are included directly in the training process.
The paper compares LaCy with several alternatives, including loss based methods, Rho-1, and LLM judge based annotations. The results show that LaCy improves factual accuracy, reduces the amount of factual knowledge stored in model parameters, and achieves these improvements at a lower annotation cost than many competing approaches.

**Audience:**

Yes

**Audience Explanation:**

The paper addresses an active research question related to small language models, retrieval augmented systems, model cascades, and agentic AI. The central finding that delegation decisions benefit from signals beyond cross entropy loss is relevant to researchers working on efficient pretraining, factuality, retrieval augmentation, and tool use.
The paper also provides evidence that lightweight factuality signals can outperform more expensive annotation approaches while achieving improved factual performance. Even researchers who may not adopt the specific spaCy based implementation can benefit from the broader insight that delegation decisions should incorporate domain specific signals beyond loss alone.

**Broader Impact Concerns:**

No significant ethical or broader impact concerns were identified beyond those typically llm training and deployment.

**Claims And Evidence:**

Yes

**Claims Explanation:**

The paper's key claims are well supported through a thorough experimental study. The proposed method, training procedure, baselines, and evaluation metrics are described clearly, making it easy to understand how the results were obtained. The authors also conduct ablation experiments that separately evaluate the impact of the loss based signal and the factual annotations, providing strong evidence that loss alone is not sufficient for determining when an SLM should delegate.
LaCy is evaluated against several competing methods, including loss based approaches, Rho-1, and LLM judge generated annotations. Across multiple metrics such as FactScore, factual leakage, validation loss, and additional ablation analyses, the results consistently favor the proposed approach. The paper further strengthens its evaluation by examining computational costs and openly discussing the limitations and applicability of the method.

**Requested Changes:**

The following suggestions would strengthen the paper but are not critical for acceptance:

1. Improve figure placement and presentation. Several figures and graphs are separated from the text that discusses them, which occasionally interrupts the reading flow.
2. Provide additional discussion comparing spaCy with alternative factuality or entity annotation approaches. While the paper motivates the use of spaCy, a brief discussion of alternative choices would further justify the design decision.
3. Consider releasing code and preprocessing scripts to further improve reproducibility. The appendix contains substantial implementation details, but a public implementation would make reproduction easier for the community.
4. Expand the discussion of potential domain transfer. The paper acknowledges that spaCy based factual annotations may be particularly suited to Wikipedia style data. Additional discussion of how the approach might generalize to other domains would strengthen the broader impact of the work.

---

> ### Author Response · Authors · 2026-06-10
> **Rebuttal**
>
> Dear Reviewer,
>
> Thank you very much for taking the time to evaluate our manuscript and for your constructive suggestions. We have uploaded a revised version of the paper that we hope addresses your comments. All newly added or substantially revised material is marked in blue. In particular, we made the following changes:
>
> 1. **Figure placement and presentation.** We reorganized the placement of figures so that each figure now appears on the same page as, or within one page of, its first reference in the text.
>
> 2. **Alternative factual annotation methods.** In the section discussing our spaCy-based implementation (page 19), we added a discussion of alternative approaches to factual annotation, including other NLP-based methods, pretrained transformer encoders with token-classification heads, and LLM judges.
>
> 3. **Reproducibility.** In Appendix B (pages 25–26), we added pseudocode for both train-time delegation and test-time calling, including the dynamic call-rate tracking procedure. We hope these additions further clarify the implementation and make the method easier to reproduce.
>
> 4. **Domain transfer.** We restructured and expanded the discussion of domain dependence, now presented in Appendix E (page 32). The revised discussion explains in greater detail:
>
>    * which properties of Wikipedia-style data make it particularly suitable for studying delegation
>    * how the central problem formulation and main the components of LaCy may transfer to other domains.
>
> We hope that these revisions address your concerns and improve the clarity and completeness of the paper. Thank you again for your careful reading and helpful feedback.

---

> > ### Comment · Reviewer_bfXs · 2026-06-14
> > **Revision Addresses Requested Changes**
> >
> > Thank you to the authors for the thorough revision and detailed response. The requested changes have been adequately addressed.
> >
> > The revised manuscript improves figure placement and presentation, includes additional discussion of alternative factual annotation approaches, expands the discussion of domain transfer, and provides further implementation details that improve reproducibility.
> >
> > I appreciate the authors' efforts in strengthening the paper and have no further concerns. The revision satisfactorily addresses my comments.

---

### Review · Reviewer_12Cj · 2026-06-18

**Summary Of Contributions:**

This paper investigates which tokens a small language model should learn to generate when equipped with external assistance, and which tokens should be delegated to a stronger model. The authors point out that relying solely on token-level loss to decide delegation is insufficient, as high-loss tokens do not necessarily lead to factual or semantic errors. Based on this observation, they propose LaCy, which uses both loss and a lightweight spaCy-based factuality signal to determine which tokens should be trained as CALL.  Experiments show that, on Wikipedia-style fact generation tasks, LaCy achieves higher factuality a ccuracy than baselines such as loss-based methods, Rho-1, and LLM-judge, while also incurring lower annotation cost. The strengths are that the problem is important, the approach is simple, the experiments are relatively thorough, the results are stable, and the cost is low. The weaknesses are that the core acceptability analysis is limited in scale, human verification of spaCy-based annotation quality is scarce, and the experiments are primarily focused on the Wikipedia-style setting, leaving cross-domain conclusions insufficiently supported.

**Additional Comments:**

This paper is generally well-written, with a clear core intuition and a relatively simple method design. I largely agree with the paper's most important insight: in the delegation learning setting for small language models, deciding which tokens should be learned and which should be delegated cannot rely solely on loss — instead, one should consider whether the token would lead to an unacceptable continuation. The authors also honestly acknowledge that spaCy is only an approximation in the current scenario, not a general-purpose solution. My main suggestions are not to change the method itself, but to further strengthen the evidence supporting the paper's core conceptual claims and to better align the claimed scope of the paper with the experimental support. If these aspects are improved, I believe this will be a valuable piece of work.

**Audience:**

Yes

**Audience Explanation:**

I believe a subset of the TMLR audience will find this work interesting, especially researchers working on small language models, model cascades, factual generation, selective prediction, tool use, and retrieval-augmented generation. The key insight of the paper is not just a concrete method but also an instructive criterion: deciding whether a token should be delegated depends not only on whether it is hard to predict, but also on whether it would lead to unacceptable factual or semantic errors. This perspective has the potential to inspire future research in this area.

**Broader Impact Concerns:**

I do not see any significant ethical risks beyond those considered normal in this line of work. The paper aims to reduce factual errors in small language model generation and improve reliability through a delegation mechanism, which is broadly positive. However, this approach does not eliminate risks but rather shifts some of the risk to the external cascaded model; therefore, the factual reliability of the final system still depends on the quality of the invoked model. I suggest the authors briefly acknowledge this point.

**Claims And Evidence:**

Yes

**Claims Explanation:**

The paper's main practical conclusion holds, in the specific scenario it studies, LaCy performs better than the compared methods.  It's just that the broader theoretical conclusions could be framed more modestly.

**Requested Changes:**

Here is the translation of your six suggestions into English:

First, I suggest that the paper more explicitly and earlier narrows the scope of its claims. The current experiments most strongly support conclusions drawn from the Wikipedia-style factual generation scenario, rather than general conclusions across all domains.

Second, if possible, I recommend strengthening the experimental support for the acceptability analysis in Section 3. This analysis currently appears to be based on a relatively small sample. If the authors could repeat the analysis across multiple batches or report a range of variability, it would make the core argument more solid.

Third, I suggest either strengthening or more clearly delineating the limitations of the spaCy labeling quality verification. The current comparison with human annotations is based on a small sample; while the results are informative, they are not sufficient to fully support strong generalizability claims.

Fourth, if space permits, I recommend including results with multiple random seeds, error margins, or at least a statement about result stability. This would increase readers' confidence in the robustness of the reported gains.

Fifth, I suggest providing further explanation regarding the fairness of the "alignment by the number of effectively supervised tokens" design in the experiments, as this affects how readers interpret several of the ablation results.

Sixth, even if no new domain experiments are added, I suggest that the authors discuss more concretely which types of tasks are suitable for a token-level acceptability approximation and which types may not be suitable.

---

> ### Author Response · Authors · 2026-07-03
> **Rebuttal**
>
> Thank you for your thoughtful and detailed feedback. We address your concerns below.
>
>
> ### 1. Scope
> Thank you for your suggestion to make our scope more visible. We have added clear statements of our findings mainly applying to wikipedia in our abstract and introduction (see also the text highlighted in blue), where we also link to our extended discussion on domain dependence (Appendix E). We hope that this further clarifies our scope.
>
>
> ### 2. Extended Acceptability Analysis
> We have now extended our acceptability analysis to 100k tokens and updated our figures (Figure 2 and Appendix F.1). The new plot looks very similar, and supports all our original conclusions. Our original plots already report Standard Errors: the very thin bands around the curves.
>
> ### 3. spaCy Labeling Quality Verification
> We have added an explicit caveat to Appendix D noting that, due to the limited scale and the use of author-produced annotations, these results should be taken as directional (highlighted in blue). We agree that small-scale human annotation alone cannot fully support strong generalizability claims. The most meaningful evidence for annotation quality at scale is therefore the downstream FactScore result: the spaCy-only model achieves comparable FactScore to the LLM judge model (Figure 6), demonstrating that spaCy annotations are practically effective as a factuality signal in our Wikipedia setting, despite their simplicity.
>
>
> ### 4. Error Margins and Result Stability
> To assess result stability, we conducted a bootstrap analysis for our main experiments both for FactScore and fact leakage. We use 10k bootstrap samples to compute 95% confidence intervals. We report symmetric uncertainties using the larger one of the upper and lower deviations. LaCy's confidence intervals are well separated from the baseline across all settings. For comparisons with LLM judge and Rho-1, individual intervals overlap at 334m given the modest evaluation set (183 entities), but the ordering consistently favours LaCy across both model sizes and all metrics, supporting the reliability of the reported gains.
>
> | Method          | Model Size | FactScore (%)    | Fact Leakage (%)  |
> |-----------------|------------|------------------|-------------------|
> | Baseline        | 334m       | 16.14 ± 1.59     | 18.45 ± 0.99      |
> | Baseline        | 1.3B       | 16.27 ± 1.70     | 17.61 ± 0.93      |
> | LaCy            | 334m       | 22.71 ± 2.03     | 14.67 ± 0.95      |
> | LaCy            | 1.3B       | 22.03 ± 2.20     | 9.24 ± 0.80       |
> | LLM judge       | 334m       | 20.97 ± 2.15     | 14.25 ± 0.92      |
> | LLM judge       | 1.3B       | 19.51 ± 1.84     | 8.71 ± 0.81       |
> | Rho-1           | 334m       | 21.63 ± 1.96     | 15.91 ± 0.94      |
> | Rho-1           | 1.3B       | 21.35 ± 1.93     | 21.21 ± 1.02      |
> | Loss-based calls| 334m       | 19.44 ± 1.98     | 19.55 ± 1.06      |
> | Loss-based calls| 1.3B       | 19.14 ± 1.81     | 16.54 ± 0.95      |
>
>
> Our experimental results use a single random seed for pre-training, because we decided to allocate our available compute to ablations on our design choices. We find that our results are robust across all settings and consistently favour LaCy. We followed prior work (LMLM, Rho-1) in running pre-training runs with a single seed.
>
> ### 5. Choice to equalize the number of effectively supervised tokens
> Thank you for your suggestion. In the main paper (Section 5.1), we now emphasize that we equalize the number of effectively supervised tokens. In Appendix F.5.1 (Fig 14), we also equalize all processed tokens. Our results are consistent across these two settings. The slight changes only concern some ablations, and are now clearly explained in Section 5.4.
>
>
> ### 6. Domain Dependence
> In response to your suggestion, Appendix E discusses concretely which types of tasks are suitable for a token-level acceptability approximation and which may not be. Wikipedia-style generation is particularly suitable: factuality violations are localised to specific tokens or short token sequences (names, dates, entities), facts are measurable at fine granularity, and there exists an established body of evaluation metrics, such as FactScore, that we make use of. In reasoning-heavy domains such as mathematics or code, correctness depends on global compositional structure rather than localised token-level facts, making token-level acceptability a less natural approximation. We additionally added an explicit pointer to this discussion in the introduction (highlighted in blue).
>
>
> ### 7.  Risk of hallucinations of the cascade partner
> Thank you for your suggestion. We have added your point into our Discussion (see our added sentences highlighted in blue).
>
>
> Thank you again for dedicating your time and expertise to reviewing our manuscript. We hope this addresses your concerns. Please do not hesitate to follow up if anything remains unclear.

---

### Review · Reviewer_6ycY · 2026-06-26

**Summary Of Contributions:**

This paper primarily investigates the challenge that, due to the limited parameter capacity of small language models (SLMs), it is difficult to internalize all factual knowledge into their parameters. Therefore, it is necessary to rationally determine which tokens are learnable and which should prompt the small model to output a \<CALL\> token to delegate the generation to external models or knowledge sources. The authors propose LaCy, which utilizes a spaCy parser to identify factual information and combines it with loss signals to determine which tokens the SLM should learn.

**Strengths:**

1. The paper reveals that tokens with high loss are not necessarily factually incorrect; they might simply lack an exact match with the ground-truth tokens in the pre-training corpus. Therefore, relying solely on loss to determine token learnability is insufficient. The authors further propose that the decision of whether a token should be learned must integrate both the loss signal and a factual assessment.

2. By training on the dwiki dataset, the authors demonstrate that LaCy achieves a higher FactScore compared to the baselines.

**Weaknesses:**

1. The overall approach is relatively straightforward and lacks algorithmic improvements.

2. The applicability of this method is primarily limited to SLMs with smaller parameter counts. Furthermore, it still relies on the assistance of external resources (such as larger-scale LMs) and cannot function independently.

**Audience:**

Yes

**Audience Explanation:**

By integrating loss signals and syntactic parsing, the paper proposes the LaCy model, which successfully learns when to make predictions autonomously and when to request assistance. This provides an effective solution for enhancing the performance of small-parameter models and improves parameter utilization.

**Broader Impact Concerns:**

No.

**Claims And Evidence:**

Yes

**Claims Explanation:**

The paper primarily trains a 334M model using the dwiki dataset and compares it against methods such as Loss-based calls, LLM judge, and Rho-1. The results indicate that its FactScore is consistently higher than existing baselines. Several ablation studies were also conducted, demonstrating that under the "spaCy only" condition, both the FactScore and factual leakage are worse. Furthermore, when considering overall performance and overhead, LaCy proves to be the most balanced method. The experimental results generally support the authors' claims.

**Requested Changes:**

1. LaCy currently determines whether to output a \<CALL\> token at the token level. However, some facts are presented in the form of phrases or short sentences. Would relying solely on token-level judgment lead to inaccurate factual segmentation?

---

> ### Author Response · Authors · 2026-07-03
> **Rebuttal**
>
> We sincerely thank you for dedicating their time and expertise for reviewing our manuscript.  We have updated our manuscript with newly added or substantially revised information marked in blue. We address your concerns below.
>
>
> ## No algorithmic improvements
>
>
> We appreciate this observation, and we agree that LaCy's contribution is not algorithmic in nature. Our method design is intentionally simple. We view this simplicity as a feature: LaCy outperforms more algorithmically complex baselines (LLM judge requires expensive LLM-based annotation, Rho-1 requires loading and querying a reference model online during training) while being cheaper and
> faster to apply. Our core claim is conceptual: loss, the sole selection signal used by the popular Rho-1 and RHO-LOSS methods, **alone** is insufficient for token delegation, and lightweight domain-specific signals can improve the delegation signal. LaCy serves as a controlled and practical instantiation of this principle, not as a claim to be the most sophisticated possible method.
>
>
>
>
> ## Limited to SLMs, requires external resources
>
>
> All methods in this field (e.g. Rho-1, LLM judge) intend to study what tokens can be learned under low capacity, thus they operate on SLMs and have some sort of partner that handles the CALL tokens. Cascade models are a simple example of this.
>
>
>
>
> ## Would relying solely on token-level judgment lead to inaccurate factual segmentation?
>
>
> To answer this question, we conducted a statistical analysis on the 334m LaCy-trained model’s biography generations used for FactScore. This new analysis can now be found in our 'Appendix F.5.3 Analysis of How LaCy Handles Multi-token Facts' (highlighted in blue).  We used our spaCy parser to annotate the generated text for factuality. This gives us a measure of ground truth factuality for the model’s generation: ideally, most facts in the generation should have come from the cascade partner.
>
> We specifically examined multi-token facts, which are facts flagged by spaCy that map to multiple tokens. We then counted, for every multi-token fact, whether LaCy called on one or multiple tokens corresponding to the fact. Since some facts may contain less truth-critical tokens, such as  ‘in’ and ‘the’ in ‘actress in the West End theatre’, we do not require the SLM to always call on every single token within the fact.
> We find that 56.93% of multi-token facts encountered multiple calls, while 30.52% encountered only one, and 12.55% encountered zero calls. Specific examples of multi-token calls can be seen in Figure 4 in the paper (e.g. Adventures of Robin Hood).
>
> Figure 4 in the paper illustrates that LaCy does not need to call on all factual tokens: for example, LaCy knows “Hood” after seeing “Robin”, in accordance with its training to delegate high-loss facts only. In addition, part of these inaccuracies can step from cases where the generated sequence could be followed by a fact, e.g. “Errol Flynn appeared in”, but the cascade partner may return a non-factual token, e.g. “more”.
>
>
> Thank you again for your careful read and constructive comments. We hope this addresses your concerns. Please do not hesitate to follow up if anything remains unclear.

---

### Review · Reviewer_5q3t · 2026-06-26

**Summary Of Contributions:**

This paper proposes LaCy, a pretraining-time token delegation method for small language models to improve model efficiency and reduce factual hallucination of SLMs. The goal is to identify tokens that an SLM should avoid learning parametrically and instead delegate through a <CALL> token.

The paper’s motivation is that loss alone is insufficient for delegation because it does not distinguish between harmful factual mistakes and harmless alternative continuations. LaCy addresses this by using linguistic factuality information to refine loss-based token selection.

LaCy innovatively incorporates non-loss-based, traditional NLP derived signals for token selection. Token loss is used to identify tokens that are difficult for the current SLM to predict. The NLP algorithm spaCy was introduced to compute factuality annotations, which are used to identify factual or truth-critical token positions. By combining these signals, LaCy delegates high-loss factual tokens rather than all high-loss tokens or all factual tokens.

The paper evaluates LaCy in a Wikipedia-style SLM cascade setting. The results show improved FactScore in biography generation, reduced fact leakage when retrieval is disabled, and comparable NLU performance relative to baseline and other delegation methods.

With factual annotations derived from spaCy (a non-LLM based algorithm), LaCy is cheaper to label than LLM-judge methods and avoids the reference-model overhead of Rho-style approaches.

**Additional Comments:**

It would be helpful if the authors included or discussed stronger hybrid baselines, such as combining SLM loss with LLM-judge factuality annotations, or combining factuality filtering with Rho-style scores. This would help clarify whether LaCy’s improvement comes specifically from the spaCy-based signal, or more generally from combining factuality and difficulty signals.

**Audience:**

Yes

**Audience Explanation:**

Yes. The paper addresses a emerging question for SLM retrieval-augmented generation and model cascades for reducing factual hallucinations: how to decide which tokens should be learned parametrically and which should be delegated to an external source.

This question is relevant to researchers working on factuality, efficient pretraining, uncertainty-based delegation, and tool-using language models especially for SLM applied in edge computing.

The main proposals loss alone is not sufficient for selecting delegation targets, and that factuality-aware token selection can improve SLM cascades—should be of interest to part of the TMLR audience. The proposed method is practically appealing because it improves over several token-selection baselines while using cheaper CPU based annotations than LLM-judge or reference-model-based methods.

**Broader Impact Concerns:**

I do not see major broader impact concerns specific to this work. The method aims to improve factual reliability in small language models by encouraging them to delegate difficult factual tokens to reduce factual hallucination.

**Claims And Evidence:**

Yes

**Claims Explanation:**

This paper provides reasonably clear evidence for its several main claims.

To show that loss alone is insufficient for deciding which tokens should be delegated. The analysis used a dwiki validation batch to show that loss predicts exact-token mismatch, but not whether the mismatch is semantically harmful. It provided evidence that high-loss factual tokens are much more likely to produce unacceptable continuations, while many high-loss non-factual tokens remain acceptable.

The evidence for LaCy's performance on factual score and factual leakage is generally positive.
* In the evaluated Wikipedia-style SLM cascade setting, LaCy improves FactScore, reduces fact leakage when retrieval is disabled while maintaining comparable NLU performance.
* The experiment compared with other approaches including loss based, LLM-judge based and Rho score based to show that LaCy performs better both in fact score and fact leakage.
* The additional ablations support the design choice of combining spaCy-based factuality annotations with loss, with spaCy-only and loss-only variants performing weaker.

There could  strongest comparison would include hybrid baselines that combine the missing signals, such as LLM-judge annotations filtered by SLM loss or Rho/reference-model scores filtered by factuality. Without these comparisons, the results show that LaCy is better than the tested baselines, but not necessarily that spaCy is the best factuality signal.

The empirical evidence would be stronger with additional experiment results on additional factual QA / fact leakage datasets. The current main evaluation is centered on Wikipedia-style biography generation, with fact leakage measured on BigBench QA Wikidata and long-tail PopQA. Additional evaluations on larger factual QA datasets such as Natural Questions-Open would provide stronger support than relying mainly on bootstrap uncertainty over the existing small evaluation set.

Overall, the claims are supported: LaCy is an effective and cheap method for improving token delegation in the paper’s Wikipedia-style SLM cascade setting.

**Requested Changes:**

1. Clarify the relative contribution of the spaCy and loss components.
LaCy combines a factuality signal together with a high-loss signal. it would be helpful to show some quantitative analysis on which component is driving the final token selection more. For example, the authors could provide an overlap analysis between LaCy-selected tokens, spaCy-only selected tokens, and loss-only selected tokens. Similar to a precision/recall-style breakdown would clarify how many LaCy calls are mainly contributed from the spaCy factuality filter, how many are mainly driven by high loss, and which tokens would be added or removed if only one component were used.

2. Provide task-specific validation of the spaCy-based factuality labels.
While spaCy already has evaluations for standard NLP tasks such as NER, LaCy relies on spaCy annotations and custom heuristics as a proxy for factual or truth-critical token positions. It would therefore be useful to include a small task-specific validation of these labels in the paper’s Wikipedia setting. Even a qualitative analysis to show that spaCy has similar performance on wikipedia settings would be sufficient.

---

> ### Author Response · Authors · 2026-07-03
> **Rebuttal**
>
> We sincerely thank the reviewer for dedicating their time and expertise for reviewing our manuscript. We have updated our manuscript with newly added or substantially revised information marked in blue. Below we address your concerns.
>
> ## Additional Fact Leakage datasets
>
> Thank you for your suggestion. We have looked into Natural Questions-Open. However, we find that these questions are too hard for our model scale, with pretraining only on Wikipedia. We already see this problem on TriviaQA, on which we now repeated our fact leakage experiments. Even baseline models achieved under 3% accuracy. We believe that this range is not appropriate to test fact leakage, as meaningful deviations cannot be distinguished from noise.
>
> We believe BigBench QA Wikidata and PopQA (setup described in Appendix C, results in Figures 3, 6, 10-16 ) remain the most suitable benchmarks for our setting, as both are grounded in Wikipedia-style knowledge and yield accuracy ranges where differences between methods are statistically interpretable.
>
>
> ## Overlap analysis between LaCy-selected tokens, spaCy-only selected tokens, and loss-only selected tokens.
>
> We now conducted an overlap analysis of tokens selected by multiple selection methods, including hybrid models suggested by the reviewer. Our new analysis is in Appendix F.5.2 Analysis of Token Selection Overlap, and is also summarized in the main text in Section 5.4 (both in blue).
>
> Our analysis scores 1.2B tokens from the dwiki validation set. For each selection method, we record the selection mask M and compare to the LaCy mask to get the overlap | M \cap LaCy | / |LaCy| (similar to precision/recall, see the paper for more details). We have computed these values for a 10% delegation budget (see Appendix F.5.2 for results for 15%).
>
> ### LaCy components
> Selecting highest-loss tokens achieves larger overlap. Our ablation on spaCy only in Figure 6 shows that spaCy only is on-par with LLM judge on FactScore. Together, these findings show that:
> * loss signal is essential, but not the only driver of LaCy selection
> * in terms of the quality of factuality signal, LLM judge and spaCy are comparable, with spaCy being cheaper
>
> ### Hybrid models
> **spaCy + rholoss** achieves a 81.3% overlap with LaCy. This shows that swapping loss for rholoss as the difficulty metric barely changes which tokens are selected. Hence, the spaCy factuality filter is the dominant driver of LaCy's selection, not the specific choice of difficulty signal.
>
>
> **LLM judge-based hybrids** produce substantially different selections (41.0% and 31.0% overlap). This is a structural consequence of LLM judge annotating only ~15% of tokens as factual, compared to spaCy's ~27%. The smaller factual pool leaves less room for loss / rholoss to refine the selection, making these methods more constrained by the factuality mask than by the difficulty signal. We therefore expect LLM judge-based hybrids to behave closer to LLM judge than to LaCy, consistent with the ablation results in Figure 6.
>
> | Method                | Overlap |
>   |-----------------------|---------|
>   | spaCy only            | 36.4%   |
>   | 10% highest loss      | 68.7%   |
>   | spaCy + rholoss       | 81.3%   |
>   | LLM judge             | 29.5%   |
>   | LLM judge + loss      | 41.0%   |
>   | LLM judge + rholoss   | 31.0%   |
>
>
> ### Qualitative Analysis
> In Appendix F.5.2, we supply examples of delegated tokens in the biography of Burke Marshall in the updated manuscript, for both 15% and 10% budgets. The main patterns we observe are:
> * at 15% budget, the LaCy and spaCy + rholoss selected sets are equal
> * LaCy or factuality + loss methods tend to delegate the hardest parts of dates, e.g. 2, 0, 3 in 2003 and 3 in 1943.
> * high loss only delegates many space tokens
>
>
> ## SpaCy Annotation Quality Analysis
> Thank you for raising this. We provide this validation in Appendix D, where we compare spaCy, LLM judge (LMLM), and human annotations on two randomly selected Wikipedia training examples.
>
> Against the human annotation as ground truth, spaCy achieves F1 scores of 75.9% and 74.7% on the two examples, outperforming LLM judge (46.1% and 66.5% respectively). This holds despite the fact that our spaCy pipeline incorporates autoregressive, model-specific rules (e.g. flagging only first occurrences of factual tokens) that diverge from standard human annotation conventions.
> Beyond this direct, qualitative comparison, the downstream FactScore results provide indirect validation at scale: the spaCy-only model achieves comparable FactScore to the LLM judge model (Figure 6), despite using a cheaper annotation.
>
> In response to your concern, we have also added a paragraph in Section A.2.1 discussing alternative approaches to spaCy-based annotation and their trade-offs, to better situate our design choice.
>
> We hope this addresses your concerns, and we thank you again for the careful and constructive review. Please do not hesitate to follow up if anything remains unclear.